# High-dimensional Bayesian Optimization via Condensing-Expansion Projection

## Abstract

In high-dimensional settings, Bayesian optimization (BO) can be expensive and infeasible. The random embedding Bayesian optimization algorithm is commonly used to address high-dimensional BO challenges. However, this method relies on the effective subspace assumption on the optimization problem's objective function, which limits its applicability. In this paper, we introduce Condensing-Expansion Projection Bayesian optimization (CEPBO), a novel random projection-based approach for high-dimensional BO that does not rely on the effective subspace assumption. The approach is both simple to implement and highly practical. We present two algorithms based on different random projection matrices: the Gaussian projection matrix and the hashing projection matrix. Experimental results demonstrate that both algorithms outperform existing random embedding-based algorithms in most cases, achieving superior performance on high-dimensional BO problems. The code is available in `https://anonymous.4open.science/r/CEPBO-14429`.

## 1 Introduction

For many optimization problems, the objective function $f$ lacks a closed-form expression, and gradient information is often unavailable, leading to what we are generally referred to as black-box functions (Jones et al., 1998; Snoek et al., 2012; Shahriari et al., 2015). Bayesian optimization (BO) is an efficient method for solving such optimization problems by modeling the unknown objective function through a probabilistic surrogate model, typically a Gaussian Process. The BO routine is a sequential search algorithm where each iteration involves estimating the surrogate model from available data and then maximizing an acquisition function to determine which point should be evaluated next. As the input space dimension $D$ increases, typically $D \geq 10$, BO encounters the so-called 'curse of dimensionality' (Bellman, 1966). This phenomenon refers to the exponential increase in the number of evaluations required to cover the input space as the dimensionality increases (Wang et al., 2016).

To address the issue, numerous studies have proposed high-dimensional BO algorithms (Wang et al., 2016; Chen et al., 2012; Binois et al., 2015; 2020; Nayebi et al., 2019; Letham et al., 2020) that typically translate high-dimensional optimizations into low-dimensional ones by various techniques, and search the new point in the low-dimensional space. However, these approaches can become inefficient when the maximum over the high-dimensional space cannot be well approximated by the maximum over the low-dimensional space.

In this paper, we introduce a novel search strategy in high-dimensional BO problems called the Condense-Expansion Projection (CEP) technique, which is both simple to implement and highly practical. In each iteration of the sequential search, the CEP technique generates a random projection matrix $\mathbf{A} \in \mathbb{R}^{d \times D}$, where $d \ll D$, to project the available data from the high-dimensional space to the low-dimensional embedding space by multiplying them with $\mathbf{A}$. It estimates the surrogate model and searches for the next point to evaluate in the low-dimensional embedding space. Subsequently, it projects the searched data point back to the high-dimensional space by multiplying it with $\mathbf{A}^\top$ for evaluation in the original space.

We employ two distinct random projection matrices to generate the projection matrix $\mathbf{A}$: the Gaussian projection matrix (Dasgupta & Gupta, 2002) and the hashing projection matrix (Rokhlin & Tygert, 2008; Boutsidis & Gittens, 2013). We show that the CEP approach preserves GP consistency in both the projection

from the high-dimensional space to the low-dimensional embedding space and the reverse projection back to the high-dimensional space. We then apply the CEP technique to two algorithms, REMBO (Wang et al., 2016) and HeSBO (Nayebi et al., 2019), resulting in the development of CEP-REMBO and CEP-HeSBO. Our experimental results, comprising comprehensive simulation studies and analysis of four real-world datasets, demonstrate that both algorithms generally outperform existing random embedding-based algorithms, showcasting the superior performance of the CEP technique on high-dimensional BO problems.

## 2 Related Work

There is a substantial body of literature on high-dimensional BO. The most closely related approach is REMBO (Wang et al., 2016) by fitting a Gaussian Process model in a low-dimensional embedding space obtained through a Gaussian random projection matrix. This approach has been further investigated under various conditions (Binois et al., 2015; 2020; Binois, 2015; Letham et al., 2020). Nayebi et al. (2019) proposed HeSBO that utilizes a hashing projection matrix. However, these studies are based on the assumption of an effective subspace, where a small number of parameters have a significant impact on the objective function. Similar to these studies, our approach evaluates the acquisition function over the embedding space. However, unlike these studies, our approach selects the new point in the embedding space and projects it back to the original space to obtain a point in the original space. The second distinguishing aspect of our approach is that it generates a new random projection matrix in each iteration.

Aside from the embedding approach, several other techniques warrant consideration. Kandasamy et al. (2015) tackled the challenges in high-dimensional BO by assuming an additive structure for the function. Other works along the line include GPs with an additive kernel (Mutný & Krause, 2018; Wang et al., 2017) or cylindrical kernels (Oh et al., 2018). However, this approach is limited by its reliance on the assumption of the additive form of the objective function. Li et al. (2017) applied the dropout technique into high-dimensional BO to alleviate reliance on assumptions regarding limited "active" features or the additive form of the objective function. This method randomly selects subset of dimensions and optimizes variables only from these chosen dimensions via Bayesian optimization. However, it necessitates "filling-in" the variables from the left-out dimensions. The proposed "fill-in" strategy, which involves copying the value of variables from the best function value, may lead to being trapped in a local optimum, although the strategy is enhanced by mixing random values. Similarly, Kirschner et al. (2019) proposed an iterative approach that solves sub-problems of the global problem, where each sub- problem selects one-dimensional subspaces of the do- main that contain the best point so far. Eriksson & Jankowiak (2021) introduced Sparse Axis-Aligned Subspace BO, which imposes a Sparse Axis-Aligned Subspace function prior to effectively identify sparse subspaces, facilitating high-dimensional BO. However, it depends on sufficiently parsimonious surrogate models. Finally, a related work is by Hvarfner et al. (2024) enhanced vanilla BO in high dimensions by appropriately scaling the lengthscale prior of the GP kernel. We will discuss its connection to our approach in the latter section.

## 3 Method

### 3.1 Bayesian Optimization

We consider the optimization problem

$$\mathbf{x}^* = \arg\max_{\mathbf{x} \in \mathcal{X}} f(\mathbf{x}),$$

where $f$ is a black-box function and $\mathcal{X} \subset \mathbb{R}^D$ is some bounded set. BO is a form of sequential model-based optimization, where we fit a surrogate model, typically a Gaussian Process (GP) model, for $f$ that is used to identify which parameters $\mathbf{x}$ should be evaluated next. The GP surrogate model is $f \sim \mathcal{GP}\left(m(\cdot), k(\cdot, \cdot)\right)$, with a mean function $m(\cdot)$ and a kernel $k(\cdot, \cdot)$. Under the GP prior, the posterior for the value of $f(\mathbf{x})$ at any point in the space is a normal distribution with closed-form mean and variance. Using that posterior, we construct an acquisition function $\alpha(\mathbf{x})$ that specifies the utility of evaluating $f$ at $\mathbf{x}$, such as Expected Improvement (Jones et al., 1998). We find $\mathbf{x}_{\text{new}} = \arg\max_{\mathbf{x} \in \mathcal{X}} \alpha(\mathbf{x})$, and in the next iteration evaluate $f(\mathbf{x}_{\text{new}})$.

However, GPs are known to predict poorly for large dimension $D$ (Wang et al., 2016), which prevents the use of standard BO in high dimensions. A common approach to addressing this challenge is linear embeddings,

which assume the existence of a low-dimensional linear subspace that captures all of the variation of $f()$. Specifically, let $\mathbf{T}^* \in \mathbb{R}^{D \times d_e}$ be a projection matrix, whose columns form an orthonormal basis for this effective subspace. The key assumption is that there exists exists a low-dimensional representation $\mathbf{y}^* \in \mathbb{R}^{d_e}$ such that

$$f(\mathbf{x}) = f(\mathbf{T}^* \mathbf{y}^*). \tag{1}$$

This defines the effective subspace in which optimization can be performed. Wang et al. (2016) proposed a random embedding via a random matrix $\mathbf{T} \in \mathbb{R}^{d \times D}$ with each element i.i.d. $\mathcal{N}(0, 1)$. By ensuring $d \geq d_e$, this approach guarantees with probability 1 that the low effective dimensionality preserved. Therefore, instead of optimizing in the high dimensional space, REMBO optimizes the function $g(\mathbf{y}) = f(\mathbf{T}\mathbf{y})$, $\mathbf{y} \in \mathbb{R}^{d_e}$ in the lower dimensional subspace.

### 3.2 Condensing-Expansion Projection

We propose an approach called Condensing-Expansion Projection (CEP) that does not rely on the assumption in (1). The CEP framework consists of two key random linear projections: one that condenses the input space to a low-dimensional space and the other that expands it back to the original input space. The name "Condensing-Expansion Projection" captures this process of first reducing the dimensionality and then restoring it.

- **Condensing Projection:** transpose points from the original space into a reduced-dimensional embedding subspace, where the surrogate model is fitted from available data and the acquisition function is maximized to determine which point should be evaluated next.

- **Expansion Projection:** revert these points in the embedding subspace back to the original space, where the searched point is evaluated.

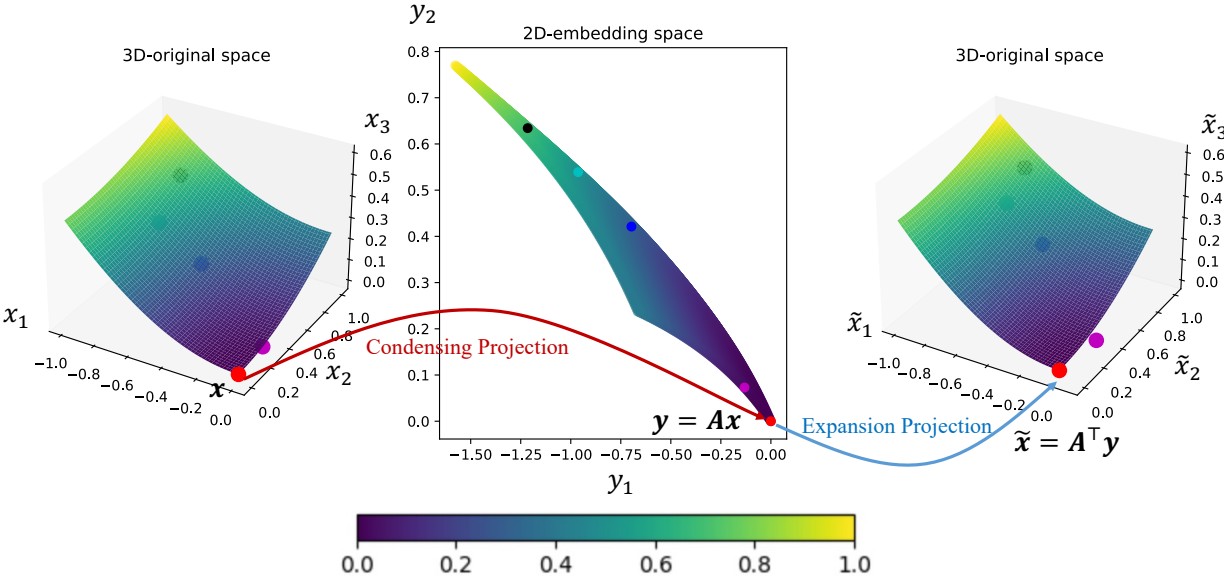

Figure 1: An illustration of CEP. The colors represent function values, as indicated by the color bar. The dimension of the original space is $D = 3$, and the dimension of the embedding subspace is $d = 2$. The five points in the original space is projected to the embedding subspace by Condensing Projection, then they are projected back to the original space by Expansion Projection. The optimal point (red dot) in the original space is still at the (approximately) optimal position after CEP.

Let us define an embedding subspace $\mathcal{Y} \subset \mathbb{R}^d$ of dimension $d$. We generate a random projection matrix $\mathbf{A} \in \mathbb{R}^{d \times D}$. Various methodologies exist for the construction of such a matrix, including the Gaussian random

matrix, sparse random matrix (Dasgupta, 2000; Bingham & Mannila, 2001), and the Subsampled Randomized Hadamard Transform (Tropp, 2011). In this paper, we utilize the Gaussian random matrix and the Hashing matrix.

Consider a point $\mathbf{x} \in \mathcal{X}$ within the original space $\mathcal{X}$. In the condensing projection, we project $\mathbf{x}$ from the original space $\mathcal{X}$ to the embedding subspace $\mathcal{Y}$ by multiplying $\mathbf{x}$ with the matrix $\mathbf{A}$, resulting in

$$\mathbf{y} = \mathbf{A}\mathbf{x} \in \mathcal{Y},$$

thereby reducing the dimension from $D$ in the original space to $d$ in the embedding subspace. In the expansion projection, we project $\mathbf{y}$ back to $\mathcal{X}$ by multiplying $\mathbf{y}$ with the transposed matrix $\mathbf{A}^\top$, expressed as as

$$\tilde{\mathbf{x}} = \mathbf{A}^\top \mathbf{y} = \mathbf{A}^\top \mathbf{A}\mathbf{x}.$$

This completes the Condensing-Expansion Projection, which can be outlined as follows: transforming a point from the original space to the embedding subspace and then restoring it back to the original space, represented as

$$\mathbf{x} \to \mathbf{y} \to \tilde{\mathbf{x}}.$$

We outline an illustration in Figure 1.

The CEP offers flexibility in selecting random projection matrix $\mathbf{A}$. In this paper, we focus on two types: Gaussian random matrices and Hashing random matrices.

**Definition 1.** *(Gaussian Random Matrix) Let $\mathbf{A} \in \mathbb{R}^{d \times D}$ be a random matrix with independent Gaussian entries. For any $1 \leq i \leq d$ and $1 \leq j \leq D$, the element $a_{ij}$ defined as*

$$a_{ij} \sim \mathcal{N}(0, 1/d).$$

**Definition 2.** *(Hashing Random Matrix) Let $\mathbf{A} \in \mathbb{R}^{d \times D}$ be a hashing random matrix. Specifically, (1) Each column of $\mathbf{A}$ has a single non-zero element that is selected at random. (2) This non-zero element has an equal probability $p = 0.5$ of being either $+1$ or $-1$.*

Assume a matrix $\mathbf{A}$ satisfying Definition 1 or Definition 2, we have

$$\mathbb{E}\left[\mathbf{A}^\top \mathbf{A}\right] = \mathbf{I}_D. \tag{2}$$

The proof of (2) is provided in the appendix 6.1. (2) represents the isometry in expectation, suggesting that, on average, the process of two linear projections preserves the information of $\mathbf{x}$.

Now we analyze the two projection steps to demonstrate their rationale in GP-based Bayesian optimization. Since the mean in the Gaussian process is typically constant, our focus is on the covariance matrix. We show that the GP kernel is approximated well in the two linear projections. First, we investigate the Condensing Projection by showing that optimization over the embedding subspace closely approximates optimization over the original space. The objective function $f()$ is fitted by a Gaussian Process model in the embedding subspace $\mathcal{Y}$:

$$f(\mathbf{y}) \sim \mathcal{GP}\left(m(\mathbf{A}\mathbf{x}), k(\mathbf{A}\mathbf{x}, \mathbf{A}\mathbf{x}')\right).$$

We establish that

$$k(\mathbf{A}\mathbf{x}, \mathbf{A}\mathbf{x}') = (1 + O_p(d^{-1/2}))k(\mathbf{x}, \mathbf{x}'). \tag{3}$$

The proof details are provided in Appendix 6.3. Hence, the Gaussian process fit in the embedding space converges to the fit in the original space.

Second, we investigate the Expansion Projection by illustrating the GP approximation in the original space, when projected back, closely approximates the GP fit in the embedding subspace. The objective function $f()$ is fitted by a Gaussian Process model over $\tilde{\mathbf{x}}$:

$$f(\tilde{\mathbf{x}}) \sim \mathcal{GP}\left(m(\mathbf{A}^\top \mathbf{y}), k(\mathbf{A}^\top \mathbf{y}, \mathbf{A}^\top \mathbf{y}')\right).$$

We show

$$k(\mathbf{A}^\top \mathbf{y}, \mathbf{A}^\top \mathbf{y}') = (1 + O_p(D^{-1/2}))k(\mathbf{y}, \mathbf{y}') \tag{4}$$

for the Gaussian Random matrix and

$$k(\mathbf{A}^\top \mathbf{y}, \mathbf{A}^\top \mathbf{y}') = (1 + O_p((D/d)^{-1/2}))k(\mathbf{y}, \mathbf{y}'), \tag{5}$$

for the Hashing Random matrix. The proof details are provided in Appendix 6.4.

Thus, by combining the approximations in (3) as well as (4) and (5), we establish the rationale for applying condensing-expansion projection in GP-based Bayesian optimization, specifically to preserve the variance function of a Gaussian process. From the analyses, we show that the CEP approach does not rely on the effective subspace assumption in (1), as the projections and ensure GP consistency, meaning that it does not depend on the structure of the GP fit.

It is worth noting that our analysis ignores the lenthscale parameters, treating them as hyperparameters. Hvarfner et al. (2024) demonstrated that appropriately scaling the lengthscale prior of the GP kernel makes vanilla BO perform well in high dimensions. A natural extension would be to implement this method of scaling lengthscale. However, a detailed investigation is needed in the future, as it is beyond the scope of this paper.

### 3.3 The CEPBO Algorithms

We employ Condensing-Expansion Projection in Bayesian Optimization, leading to the development of the Condensing-Expansion Projection Bayesian Optimization (CEPBO) algorithms. In contrast to Random Embedding algorithms, such as REMBO(Wang et al., 2016), HeSBO(Nayebi et al., 2019) and ALEBO(Letham et al., 2020), where a fixed projection matrix is employed, the CEPBO algorithms dynamically generate a new projection matrix $\mathbf{A}_t$ during each iteration $t$.

The Random Embedding algorithms keeps the random embedding fixed and rely on the effective subspace assumption to enable a deterministic Gaussian process model (Wang et al., 2016; Cartis et al., 2023), with the suggestion of using several random embeddings. Unlike these algorithms, our approach does not depend on this assumption. Instead, our analysis reveals an approximation error in the GP fit. To further reduce this approximation error, we adopt the strategy of iteratively generating a new projection matrix at each iteration.

Through Condensing Projection, which condenses available points from the original space to a new embedding subspace via multiplication with $\mathbf{A}_t$, CEPBO leverages past information to conduct BO within the embedding subspace. It determines which point to evaluate next within this subspace. Afterward, the selected point in the embedding subspace undergoes Expansion Projection, where it is projected back to the original space via multiplication with $\mathbf{A}_t^\top$. Subsequently, the objective function is then evaluated at the chosen point.

The detailed procedural flow of the algorithms is outlined in Algorithm 1. By using different random projection matrices at line 4 of the Algorithm 1, we derive two algorithms: CEP-REMBO and CEP-HeSBO. These can be regarded as enhanced versions of REMBO and HeSBO, respectively.

**Condense original space into the embedding subspace.** The core concept of employing Condensing Projection involves creating a new subspace $\mathcal{Y}$ at each iteration, where BO is subsequently performed. However, since the historical trajectories are preserved within the original space $\mathcal{X}$, the newly formed subspace must be equipped with the necessary information to enable effective BO. To tackle this issue, the primary objective of Condensing Projection is to transfer the historical trajectories from the original space $\mathcal{X}$ into an embedding subspace $\mathcal{Y}$, thereby furnishing the embedding subspace $\mathcal{Y}$ with the necessary information to facilitate BO. At the current iteration $t$, let $\mathcal{D}_{t-1}$ represent the trajectories in the original space $\mathcal{X}$, given by: $\mathcal{D}_{t-1} = \{(\mathbf{x}_1, f(\mathbf{x}_1)), (\mathbf{x}_2, f(\mathbf{x}_2)), \ldots, (\mathbf{x}_{t-1}, f(\mathbf{x}_{t-1}))\}$. During this iteration, a new projection matrix $\mathbf{A}_t$ of dimensions $\mathbb{R}^{d \times D}$ is sampled. This matrix serves as projecting the historical point from the original space $\mathcal{X}$ into a newly formed embedding subspace $\mathcal{Y}_t$: $\mathcal{D}_{t-1}^y = \{(\mathbf{A}_t \mathbf{x}_1, f(\mathbf{x}_1)), (\mathbf{A}_t \mathbf{x}_2, f(\mathbf{x}_2)), \cdots, (\mathbf{A}_t \mathbf{x}_{t-1}, f(\mathbf{x}_{t-1}))\}$.

**Optimize over the embedding subspace.** The objective $f$ is fitted by a Gaussian Process model over the embedding subspace $\mathcal{Y}_t$. From (3), we see that the Gaussian process fit in the embedding space

---

**Algorithm 1:** CEPBO Algorithms

---

**Input:** Objective $f : \mathcal{X} \to \mathbb{R}$; Acquisition criterion $\alpha$; Original dimension $D$; Embedding dimension $d$;
Initial points $t_0$; Evaluation trials $t_N$

**Output:** Best point $\mathbf{x} \in \arg\max\limits_{\mathcal{X}} f(\mathbf{x})$

**1** Uniformly sample $t_0$ points $\{\mathbf{x}_1, \mathbf{x}_2, \cdots, \mathbf{x}_{t_0}\}$ in the original space;

**2** Define $\mathcal{D}_0 = \{(\mathbf{x}_1, f(\mathbf{x}_1)), (\mathbf{x}_2, f(\mathbf{x}_2)), \ldots, (\mathbf{x}_{t_0}, f(\mathbf{x}_{t_0}))\}$;

**3 while** $t_0 + 1 \leq t \leq t_N$ **do**

**4**     Construct the projection matrix $\mathbf{A}_t \in \mathbb{R}^{d \times D}$ according to Gaussian projection matrix in the Definition 1 or hashing projection matrix in the Definition 2;

**5**     Project the points in $\mathcal{D}_{t-1}$ onto the embedding subspace $\mathcal{Y}_t$ via $\mathbf{A}_t$, obtaining the set of points in the embedding subspace $\mathcal{D}_{t-1}^y = \{(\mathbf{A}_t\mathbf{x}_1, f(\mathbf{x}_1)), (\mathbf{A}_t\mathbf{x}_2, f(\mathbf{x}_2)), \cdots, (\mathbf{A}_t\mathbf{x}_{t-1}, f(\mathbf{x}_{t-1}))\}$;

**6**     Estimate the hyperparameters $\theta_t$ of the Gaussian Process prior for the given $\mathcal{D}_{t-1}^y$;

**7**     Calculate the posterior probability of the Gaussian Process based on $\mathcal{D}_{t-1}^y$ and the estimated hyperparameters $\theta_t$.

**8**     Compute the maximum of the acquisition criterion $\alpha$, $\mathbf{y_t} \in \arg\max\limits_{\mathbf{y} \in \mathcal{Y}} \alpha(\mathbf{y} \mid \mathcal{D}_{t-1}^y)$;

**9**     Project $\mathbf{y}_t$ back to the original space via $\mathbf{A}_t^\top$, obtaining $\mathbf{x}_t = \mathbf{A}_t^\top \mathbf{y_t}$;

**10**     Update the dataset $\mathcal{D}_t = \mathcal{D}_{t-1} \cup \{(\mathbf{x}_t, f(\mathbf{x}_t))\}$, and $t = t + 1$.

**11 end**

---

converges to the fit in the original space. Within the embedding subspace $\mathcal{Y}_t$, the dataset $\mathcal{D}_{t-1}^y$ informs the estimation of the hyperparameters $\theta_t$ for the Gaussian process, and the posterior probability of the Gaussian process is computed. The acquisition function $\alpha$ (such as Expected Improvement) identifies the embedding subspace's optimal point $\mathbf{y}_t^*$ within the embedding subspace, which is represented by the equation $\mathbf{y}_t^* = \arg\max\limits_{\mathbf{y} \in \mathcal{Y}_t} \alpha(\mathbf{y} \mid \mathcal{D}_{t-1}^y)$.

**Project back and evaluate in the original space.** After searching the optimal point with the acquisition function, we need to project this point back to the original space $\mathcal{X}$ for objective function evaluation. Subsequently, we add this point to the historical trajectories. To be more specific, we use the transpose projection matrix $\mathbf{A}_t$ to map the optimal point $\mathbf{y}_t^*$ from the embedding subspace back to $\mathcal{X}$ by applying its transpose $\mathbf{A}_t^\top$, expressed as: $\tilde{\mathbf{x}}_t^* = \mathbf{A}_t^\top \mathbf{y}_t^*$. As demonstrated in (4), the Gaussian process fit in the original space, when projected back, approximates the fit in the embedding space. Subsequently, we evaluate the objective function at $\tilde{\mathbf{x}}_t^*$ within $\mathcal{X}$ to obtain $f(\tilde{\mathbf{x}}_t^*)$. This data, denoted as $(\tilde{\mathbf{x}}_t^*, f(\tilde{\mathbf{x}}_t^*))$, is then added to the historical trajectories $\mathcal{D}_{t-1}$, resulting in: $\mathcal{D}_t = \mathcal{D}_{t-1} \cup \{(\tilde{\mathbf{x}}_t^*, f(\tilde{\mathbf{x}}_t^*))\}$. This completes a full iteration cycle of the CEPBO algorithms.

### 3.4 Address the boundary issue

Our approach, akin to REMBO, encounters the issue of excessive exploration along the boundary of $\mathcal{X}$. To ensure the effective tuning of the acquisition function and to facilitate BO, it is crucial for the embedding subspace $\mathcal{Y}$ to have a bounded domain. However, random projections between the original space of dimension $D$ and the embedding subspace of dimension $d$ can lead to points exceeding domain boundaries after CEP. These exceedances occur in two scenarios: $\mathbf{y} = \mathbf{A}\mathbf{x} \notin \mathcal{Y}$, $\tilde{\mathbf{x}} = \mathbf{A}^\top \mathbf{y} \notin \mathcal{X}$.

Following the convex projection strategy (Wang et al., 2016; Binois et al., 2020; Letham et al., 2020), to mitigate the issue, we employ the convex projection of the original space, $P_\mathcal{X}$, and that of the embedding subspace, $P_\mathcal{Y}$, to rectify boundary transgressions. Specifically, the convex projection within the original space $\mathcal{X}$ is defined as follows:

$$P_\mathcal{X} : \mathbb{R}^D \to \mathbb{R}^D, \ P_\mathcal{X}(\tilde{\mathbf{x}}) = \arg\min_{\mathbf{z} \in \mathcal{X}} \|\mathbf{z} - \tilde{\mathbf{x}}\|_2.$$

Similarly, the convex projection within the embedding subspace $\mathcal{Y}$ is expressed as:

$$P_\mathcal{Y} : \mathbb{R}^d \to \mathbb{R}^d, \ P_\mathcal{Y}(\mathbf{y}) = \arg\min_{\mathbf{z} \in \mathcal{Y}} \|\mathbf{z} - \mathbf{y}\|_2.$$

Convex projection will lead to an issue where multiple distinctive values in the original space are mapped to identical boundary points within the embedding subspace, i.e., for $\mathbf{x}_1 \neq \mathbf{x}_2$ such that $f(\mathbf{x}_1) \neq f(\mathbf{x}_2)$, the equality $P_{\mathcal{Y}}(\mathbf{A}\mathbf{x}_1) = P_{\mathcal{Y}}(\mathbf{A}\mathbf{x}_2)$ holds. Moreover, the substantial disparity between the dimensions $d$ and $D$ exacerbates the likelihood of such instances. This issue can undermine the precision of Gaussian process models and consequently, diminish the efficacy of optimization. To mitigate this risk, a scaling strategy is implemented within the Condensing Projection and Expansion Projection phases to diminish the probability of such occurrences. This involves scaling the projected points $\mathbf{A}\mathbf{x}$ using a reduction factor before applying convex projection, as follows:

$$\mathbf{y} = P_{\mathcal{Y}}\left(D^{-1/2}\mathbf{A}\mathbf{x}\right).$$

In a parallel procedure, the optimal points of the acquisition function in the embedding subspace $\mathbf{y}$ undergo an inverse scaling:

$$\tilde{\mathbf{x}} = P_{\mathcal{X}}(D^{1/2}\mathbf{A}^{\top}\mathbf{y}).$$

In this context, the scaling factors $1/\sqrt{D}$ and $\sqrt{D}$ confine the scope of projection within the viable domain. These factors are verified through empirical experimentation.

## 4 Experiments

We conduct experiments to demonstrate the performance of the proposed method across various functions and real-world scenarios. In Section 4.1, we evaluate its performance on three benchmark functions. In Section 4.2, we assess it across four real-world problems. These experimental results indicate that our algorithms, CEP-REMBO and CEP-HeSBO, achieve superior results.

Because our approach, CEPBO, represents an advancement in the domain of the linear embeddings, our experiments focus on comparing it with other linear embeddings. We aim to assess the improvement achieved by applying CEP compared to REMBO (Wang et al., 2016) and HeSBO (Nayebi et al., 2019), respectively. ALEBO (Letham et al., 2020) is considered to achieve state-of-the-art performance on this class of optimization problems with a true linear subspace. Therefore, we chose REMBO (Wang et al., 2016), HeSBO (Nayebi et al., 2019), and ALEBO (Letham et al., 2020) as benchmark algorithms for our comparisons. In addition to these subspace-based algorithms, we also include two recent advanced algorithms operating in the original space, SAASBO (Eriksson & Jankowiak, 2021) and VanillaBO (Hvarfner et al., 2024), as benchmarks.

### 4.1 Numerical Results

We evaluated the performance of the algorithms using the following benchmark functions: (1) the Holder Table function, (2) the Schwefel function, and (3) the Griewank function. Each function's input space was extended to a dimensionality of $D = 100$. The Holder Table function is a two-dimensional function, meaning it has an effective dimension of 2, while BO attempts to fit it in a $D$-dimensional space. In contrast, the Schwefel and Griewank functions are defined over the entire $D$-dimensional space, with an effective dimension of $D$. See Appendix 6.6 for their definitions. The goal is to find the minimum value of these functions. The number of initialization trials for each algorithm was kept the same as the dimensionality of its embedding subspace. Each experiment is independently repeated 50 times, with 50 evaluations per experiment. To assess the performance of the CEPBO algorithm under various embedding space dimensions, we take $d = 2, 5, 10$ in the Holder Table function and $d = 2, 5, 20$ in the Schwefel and Griewank functions. Since an effective dimension for the Schwefel and Griewank functions is 100, we prioritize the larger $d$ for assessment. In these experiments, we utilize expected improvement as the acquisition function.

We report the results in Figure 2. First, we compare against subspace-based algorithms: REMBO, HeSBO, and ALEBO. For the Schwefel and Griewank functions, where embedding dimensions are smaller than the effective dimension of 100, the baseline algorithms nearly ceased functioning, settling in local optima, which is visually depicted as a flat horizontal line on the corresponding graphs. Interestingly, even in the instance of the Holder Table function, where the embedding dimension met or exceeded the effective dimension, a circumstance where the baseline algorithms typically perform well, the approached algorithms continued to show superior performance over all baselines. Comparative the performance across a range of embedding

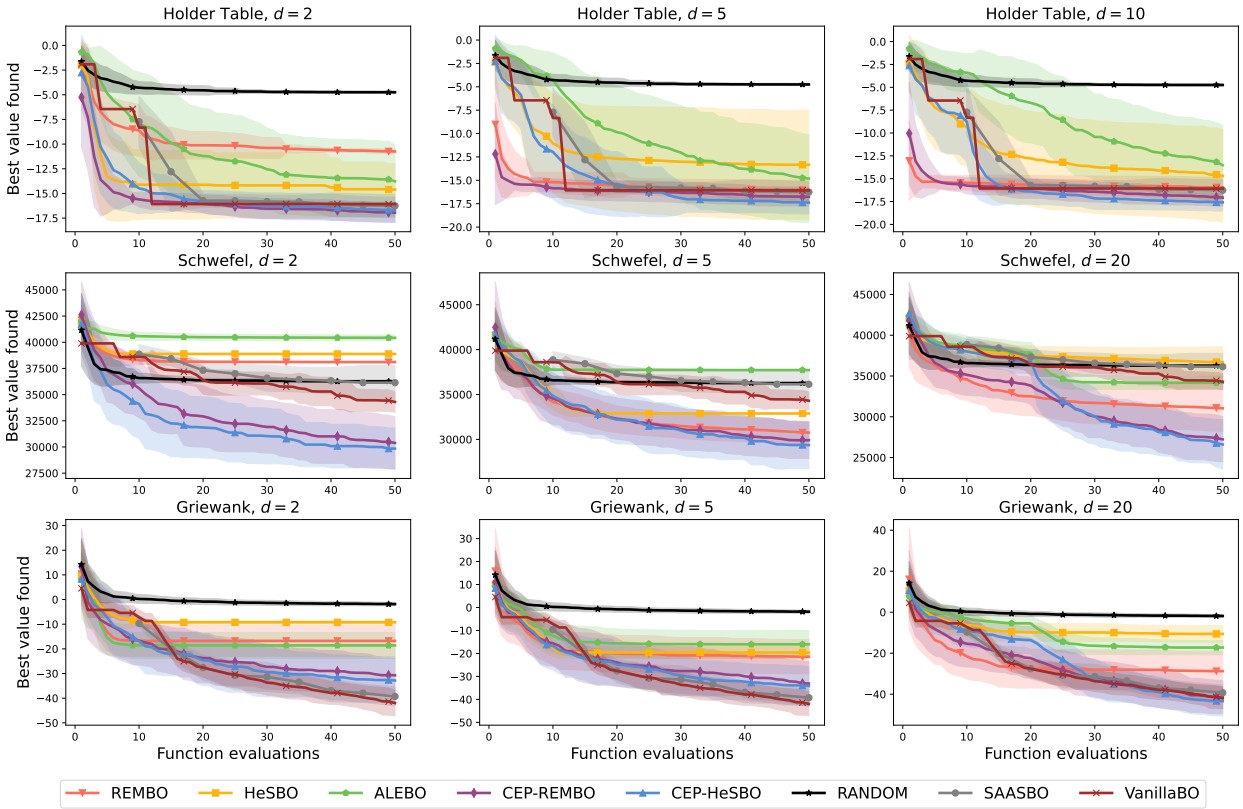

Figure 2: The results of the optimization experiments for three functions across various embedding dimensions. From top to bottom: Holder Table, Schwefel, and Griewank functions.

dimensions $d$, the performance are similar and CEP-REMBO and CEP-HeSBO consistently surpassed all baseline algorithms. Therefore, the REMBO and HeSBO algorithms experienced substantial improvement with the integration of the Condense-Expansion Projection mechanism.

Second, compared to the algorithms operating in the original space, SAASBO and VanillaBO, which outperform REMBO, HeSBO, and ALEBO, our approach achieves superior performance on the Holder and Schwefel functions. For the Griewank function, our algorithms performs slightly worse when $d = 2$ or 5, but outperform them as $d$ increases $d = 20$.

**Impact of higher dimension $D$.**

To assess the performance of the CEPBO algorithm in higher dimensions, we conducted simulations using the well-known Hartmann function. Specifically, we utilized the Hartmann function with an original space dimension of $D = 6$ and set the embedded space dimension to $d = 6$ as well. To simulate a high-dimensional environment, we expanded the original space from $D = 6$ to $D = 1000$, but in practice, only the 6-dimensional data is valid. Note that we exclude SAASBO and VanillaBO as competitors, as fitting the GP in such high=dimensional setting is costly and leads to significantly longer computation time.

The results are reported on the left in Figure 3. In this setup, ALEBO, REMBO, and HeSBO were identified as the best-performing configurations. The experimental results demonstrate that our proposed algorithm still maintains a certain level of superiority, with the CEP-REMBO algorithm being the optimal one. Additionally, the results indicate that incorporating the CEP projection mechanism can significantly improve the performance of both REMBO and HeSBO algorithms.

To realistically simulate the optimization performance of the CEPBO algorithm on an actual 1000-dimensional function, we still use the settings from Section 4.1. However, we employ an Griewank function with an effective dimension of 1000, where all 1000 dimensions have a tangible impact on the function results.

As shown on the middle in Figure 3, other algorithms, apart from CEPBO, quickly fail and become trapped in local optima. This indicates that even in extremely high dimensions, once the effective dimension of the space exceeds the embedding dimension, non-CEPBO algorithms struggle to perform. However, our projection can effectively alleviate this issue, allowing for continuous searching for optimal solutions even when using a very small embedding space.

**Robustness to noisy rewards.**

To assess the performance of the CEPBO algorithm in a noisy setting, we conducted simulations using the well-known Holder Table function, and set $d = 2$. Specifically, the function settings were the same as those outlined in Section 4.1. Furthermore, during the iterations of the Bayesian algorithm, we introduced a random normal distribution noise disturbance to the reward function, where $\epsilon \sim N(0,1)$, to simulate noise in real-world environments.

The results are illustrated on the right in Figure 3. The experimental findings indicate that our proposed algorithm retains a significant advantage, with the CEP-HeSBO algorithm demonstrating the best performance. The results suggest that incorporating the CEP projection mechanism is robust to the noisy rewards.

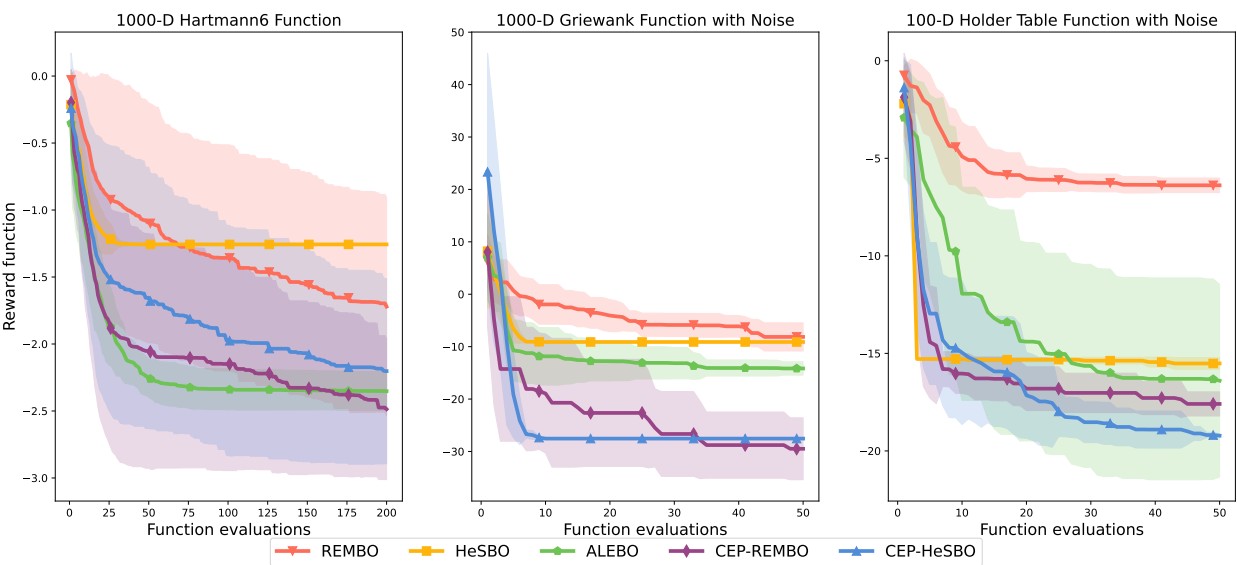

Figure 3: From left to right: the 1000-$D$ Hartmann function, 1000-$D$ Griewank function, and the Noisy Holder Table function.

## 4.2 Real-World Problems

In this section, we evaluate the CEPBO algorithm on real-world optimization problems. The test cases consist of lunar landing task in the realm of reinforcement learning with $D = 12$ (Eriksson et al., 2019), a robot pushing problem with $D = 14$ (Wang et al., 2017), a problem in neural architecture search with $D = 36$ (Letham et al., 2020), and a rover trajectory planning problem with $D = 60$ (Wang et al., 2018). Algorithmic configurations and acquisition function selections strictly adhere to the settings outlined in the original papers. For additional details, please refer to the appendix. The optimization goal is to maximize the reward function, and each experiment is independently repeated 10 times, with 500 evaluations per experiment.

**Lunar Landing.** This experiment entails the task of devising a reinforcement learning strategy for the lunar lander's control mechanism, aiming to minimize fuel consumption and proximity to the landing site while preventing a crash. The original space dimension is $D = 12$. In the first column of Figure 4, REMBO,

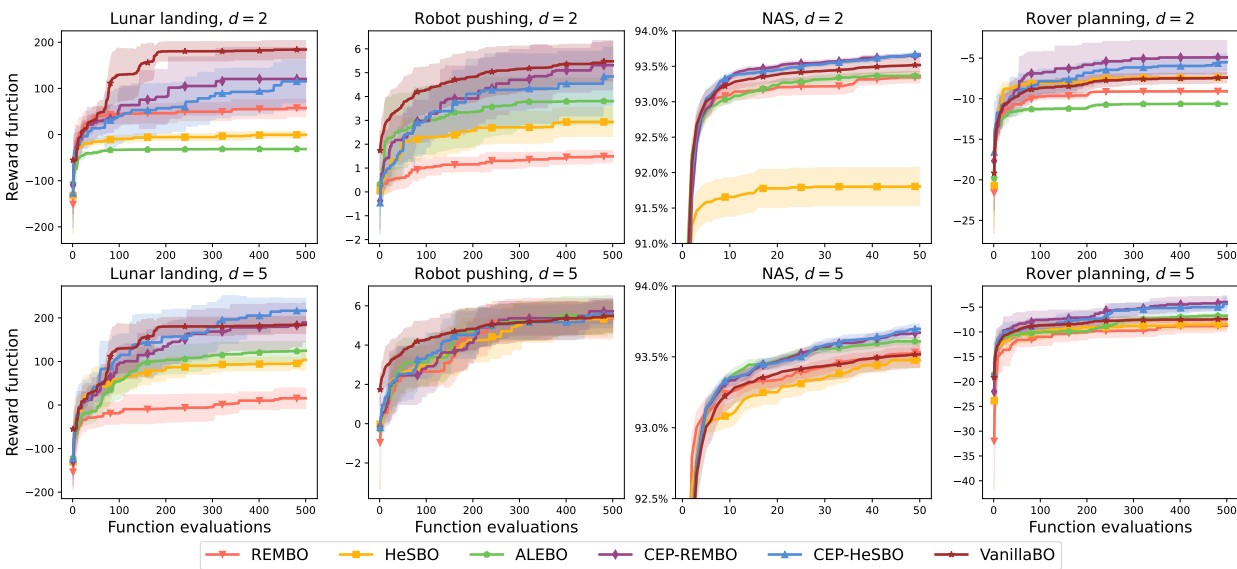

Figure 4: The results of the optimization experiments for four real-world scenarios. From left to right: Lunar landing, Robot pushing, NAS, and Rover planning.

HeSBO, and ALEBO algorithms become trapped in local optima to different extents. As the dimensionality of the embedding subspace $d$ increases from $d = 2$ to $d = 5$, notable performance improvements are observed for most algorithms, except REMBO. The introduced CEP-REMBO and CEP-HeSBO algorithms consistently demonstrate the ability to enhance and identify novel optimal resolutions. Now we compare with VanillaBO. Note that when comparing with algorithms operating in the original space, we focus on VanillaBO (Hvarfner et al., 2024) due to its strong performance in Section 4.1. Our algorithms perform slightly worse than VanillaBO when $d = 2$. However, as observed in Section 4.1, increasing to $d = 5$ allows our algorithms to outperform it.

**Robot Pushing.** This scenario involves a robotics dual-arm manipulation task where the robot's arms are controlled by adjusting 14 modifiable parameters to push two objects while tracking their movement trajectories. The original space dimension is $D = 14$. The second column of Figure 4 demonstrates that the proposed CEP-REMBO and CEP-HeSBO methods significantly outperform others when $d = 2$. When increasing to $d = 5$, all methods exhibit varying degrees of performance enhancement. As observed in Cartis et al. (2023), this suggests that for optimization issues with moderate to low dimensionalities, escalating the dimensions of the embedding subspaces can notably bolster the algorithms' efficacy. Notwithstanding these improvements, the CEP-REMBO and CEP-HeSBO methods consistently maintain their leading positions. VanillaBO performs best when $d = 2$ in this case. However, similar to the Lunar landing case, increasing to $d = 5$ improves the performance of subspace-based algorithms and allows our algorithms to outperform it.

**Neural Architecture Search (NAS).** The objective of this experiment is to identify an optimal architecture for neural networks, paralleling the methodology utilized by Letham et al. (2020). Leveraging data from the NAS-Bench-101 benchmark Ying et al. (2019), we have developed an optimization problem focused on searching for a convolutional neural network architecture characterized by 36 dimensions. The original space dimension is $D = 36$. In the third column of Figure 4, at an embedding subspace dimension of $d = 2$, the REMBO, HeSBO, and ALEBO algorithms rapidly converge to less than ideal solutions, hindering the exploration of superior neural network structures. On the contrary, the CEP-REMBO and CEP-HeSBO methods maintain the capability to persistently optimize, discovering architectures with improved accuracy. Increasing the subspace dimension to $d = 5$ reveals the ALEBO's enhanced ability to perform on par with CEP-REMBO and CEP-HeSBO methods; however, CEP-HeSBO consistently exhibits the highest performance across all conditions. In this case, VanillaBO performs worse than our algorithms when $d = 2$,

and increasing to $d = 5$ improves the performance of subspace-based algorithms, further enhancing the outperformance of our algorithms.

**Rover Trajectory Planning.** This task involves optimizing a 2D trajectory comprising of 30 pivotal points that collectively define a navigational path. The original space dimension is $D = 60$. The fourth column of Figure 4 indicates that for an embedding subspace dimension of $d = 2$, the REMBO, HeSBO, and ALEBO algorithms can not successfully converge to an advantageous reward. In contrast, the CEP-REMBO and CEPHeSBO algorithms exhibit a capacity to consistently identify superior solutions. This pattern is similarly observed when the subspace dimension is increased to $d = 5$. VanillaBO performs worse than our algorithms in this case, and the results are similar to those in the NAS case.

## 5   Conclusion

This paper proposes a Bayesian optimization framework utilizing the Condensing-Expansion Projection technique, free from reliance on the assumption of effective dimension. The primary concept involves employing projection twice within each iteration: first, projecting to an embedding subspace, and then projecting back to retain optimization information in the original high-dimensional space. Our CEP approach does not impose additional requirements on the projection matrix used, thereby significantly enhancing the applicability of the embedding-based Bayesian optimization algorithms. Two new Bayesian optimization algorithms based on Condensing-Expansion Projection are proposed: CEP-REMBO and CEP-HeSBO based on the Gaussian projection matrix and the hash-enhanced projection matrix, respectively. Empirically, this paper conducts comprehensive experiments to assess the performance of the proposed algorithms across diverse optimization scenarios. The experimental results demonstrate that the Bayesian optimization algorithms based on Condensing-Expansion Projection achieved promising performance across these optimization functions, overcoming the reliance on effective dimension.

For previous embedding-based Bayesian optimization algorithms, achieving an optimal solution requires $d$ to be greater than or equal to the true effective dimension of the optimization problem. In contrast, our algorithms do not have this requirement and perform robustly across different choices of $d$. In practice, $d$ can be viewed as a hyperparameter to be set. When selecting $d$, it is crucial to balance the approximation error from Condensing-Expansion Projection and 'curse of dimensionality' of Bayesian optimization. Empirically, our approach performs robustly with respect to the choice of $d$ when $d$ ranges from 2 to 20, and increasing $d$ can be beneficial. Setting $d = 5$ is typically effective.

Our work has several limitations that can be addressed in future studies. For instance, one limitation is the absence of an evaluation of CEP-based algorithms for optimization problems with billions of dimensions. Despite this potential, our approach lacks empirical validation, whereas REMBO has been shown to effectively address such challenges. Another limitation is that our analysis ignores the lenthscale parameters, treating them as hyperparameters. A natural extension would be to implement the method of scaling lengthscale in vanilla BO, as proposed by Hvarfner et al. (2024), which is promising direction for future work.

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

## 6 Appendix

### 6.1 Proof of (2)

*Proof.* (1) First, we consider the Gaussian Projection. Let $\mathbf{A} = (\boldsymbol{l}_1, \cdots, \boldsymbol{l}_d)^\top$, such that $\boldsymbol{l}_i = (\alpha_{i1}, \cdots, \alpha_{iD})^\top$ is a $D \times 1$ vector where each element is from $\mathcal{N}\left(0, d^{-1}\right)$ distribution. It is easy to show that, given any vector $\mathbf{x} \in \mathcal{X}$ and any vector $\mathbf{g} \in \mathbb{R}^D$,

$$\mathbb{E}(\mathbf{A}^\top \mathbf{A}) = \sum_{i=1}^d \mathbb{E}\left(\boldsymbol{l}_i \boldsymbol{l}_i^\top\right) = \mathbf{I}.$$

(2) Second, we consider the Hashing Random Projection. For the Hashing Random Matrix, we rewrite $\mathbf{A} = \mathbf{SD}$, where $\mathbf{S} \in \mathbb{R}^{d \times D}$ has each column chosen independently and uniformly from the $r$ standard basis vectors of $\mathbb{R}^d$ and $\mathbf{D} \in \mathbb{R}^{D \times D}$ is a diagonal matrix with diagonal entries chosen independently and uniformly on $b_i \in \{\pm 1\}$.

Let $\mathbf{S} = (\boldsymbol{s}_1, \cdots, \boldsymbol{s}_D)$, such that $\boldsymbol{s}_i$ is a random vector taking the vector $e_j$ for equal probability, where $e_k$ is the $k$th standard unit vector of $\mathbb{R}^d$ for $k = 1, \cdots, d$. Then $\mathbb{E}(\boldsymbol{s}_i) = d^{-1} \mathbf{1}_d$ and $\mathbb{E}(\boldsymbol{s}_i^\top \boldsymbol{s}_i) = 1$, which follow that $\mathbb{E}[(\mathbf{S}^\top \mathbf{S})_{ij}] = \mathbb{E}(\boldsymbol{s}_i^\top \boldsymbol{s}_j) = \mathbb{E}(\boldsymbol{s}_i)^\top \mathbb{E}(\boldsymbol{s}_j) = \frac{1}{d^2}$ for $i \neq j$; and $\mathbb{E}[(\mathbf{S}^\top \mathbf{S})_{ii}] = \mathbb{E}(\boldsymbol{s}_i^\top \boldsymbol{s}_i) = 1$. Obviously,

$$\mathbb{E}(\mathbf{A}^\top \mathbf{A}) = \mathbf{I}.$$

$\square$

### 6.2 Two proposition

**Proposition 1.** *Let $\mathbf{A}$ is a $d \times D$ matrix where each element is independently from $\mathcal{N}\left(0, d^{-1}\right)$ distribution, then we have*

$$\mathbb{E}\left[(\mathbf{x}^\top \mathbf{A}^\top \mathbf{A} \mathbf{x} - \mathbf{x}^\top \mathbf{x})^2\right] = 2d^{-1} \|\mathbf{x}\|^4; \tag{6}$$

$$\mathbb{E}\left[(d/D\mathbf{y}^\top \mathbf{A} \mathbf{A}^\top \mathbf{y} - \mathbf{y}^\top \mathbf{y})^2\right] = 2D^{-1} \|\mathbf{y}\|^4. \tag{7}$$

*Proof.* Let $\mathbf{A} = (\boldsymbol{l}_1, \cdots, \boldsymbol{l}_d)^\top$, such that $\boldsymbol{l}_i = (\alpha_{i1}, \cdots, \alpha_{iD})^\top$ is a $D \times 1$ vector where each element is from $\mathcal{N}\left(0, d^{-1}\right)$ distribution. (2) follows that

$$\mathbb{E}[(\mathbf{x}^\top \mathbf{A}^\top \mathbf{A} \mathbf{x} - \mathbf{x}^\top \mathbf{x})^2] = \mathbb{E}[(\mathbf{x}^\top \mathbf{A}^\top \mathbf{A} \mathbf{x})^2] - (\mathbf{x}^\top \mathbf{x})^2. \tag{8}$$

In (8), we have,

$$\mathbb{E}\left[(\mathbf{x}^\top \mathbf{A}^\top \mathbf{A} \mathbf{x})^2\right] = \mathbb{E}\left[\left(\sum_{i=1}^d \mathbf{x}^\top \boldsymbol{l}_i \boldsymbol{l}_i^\top \mathbf{x}\right)^2\right] = \mathbb{E}\left[\sum_{i=1}^d (\mathbf{x}^\top \boldsymbol{l}_i \boldsymbol{l}_i^\top \mathbf{x})^2 + \sum_{i \neq j} (\mathbf{x}^\top \boldsymbol{l}_i \boldsymbol{l}_i^\top \mathbf{x})(\mathbf{x}^\top \boldsymbol{l}_j \boldsymbol{l}_j^\top \mathbf{x})\right]$$

$$= d\mathbb{E}[(\mathbf{x}^\top \boldsymbol{l}_i \boldsymbol{l}_i^\top \mathbf{x})^2] + (d^2 - d)\left[\mathbb{E}(\mathbf{x}^\top \boldsymbol{l}_i \boldsymbol{l}_i^\top \mathbf{x})\right]^2.$$

By (17) in Lemma 1,

$$\mathbb{E}[(\mathbf{x}^\top \boldsymbol{l}_i \boldsymbol{l}_i^\top \mathbf{x})^2] = \frac{3}{d^2} \|\mathbf{x}\|^4.$$

Then we have

$$\mathbb{E}\left[(\mathbf{x}^\top \mathbf{A}^\top \mathbf{A} \mathbf{x})^2\right] = (\mathbf{x}\top\mathbf{x})^2 + \frac{2}{D} \|\mathbf{x}\|^4.$$

Therefore, (6) is proved.

Let $\mathbf{A} = (\mathbf{a}_1, \cdots, \mathbf{a}_D)$. A simple calculation shows

$$\mathbb{E}[\mathbf{A} \mathbf{A}^\top] = \sum_{i=1}^D \mathbf{a}_i \mathbf{a}_i^\top = D/d\mathbf{I}_d.$$

Then we have

$$\mathbb{E}[(d/D\mathbf{y}^\top \mathbf{A} \mathbf{A}^\top \mathbf{y} - \mathbf{y}^\top \mathbf{y})^2] = \mathbb{E}[d^2/D^2 (\mathbf{y}^\top \mathbf{A} \mathbf{A}^\top \mathbf{y})^2] - (\mathbf{y}^\top \mathbf{y})^2.$$

For the term $\mathbb{E}[(\mathbf{y}^\top \mathbf{A}\mathbf{A}^\top \mathbf{y})^2]$, we have,

$$
\begin{aligned}
\mathbb{E}\left[(\mathbf{y}^\top \mathbf{A}\mathbf{A}^\top \mathbf{y})^2\right] =& \mathbb{E}\Big[(\sum_{i=1}^{D} \mathbf{y}^\top \mathbf{a}_i \mathbf{a}_i^\top \mathbf{y})^2\Big] = \mathbb{E}\Big[\sum_{i=1}^{D}(\mathbf{y}^\top \mathbf{a}_i \mathbf{a}_i^\top \mathbf{y})^2 + \sum_{i \neq j}(\mathbf{y}^\top \mathbf{a}_i \mathbf{a}_i^\top \mathbf{y})(\mathbf{y}^\top \mathbf{a}_j \mathbf{a}_j^\top \mathbf{y})\Big] \quad (9)\\
=& D\mathbb{E}[(\mathbf{y}^\top \mathbf{a}_i \mathbf{a}_i^\top \mathbf{y})^2] + (D^2 - D)\left[\mathbb{E}(\mathbf{y}^\top \mathbf{a}_i \mathbf{a}_i^\top \mathbf{y})\right]^2.
\end{aligned}
$$

By (17) in Lemma 1,

$$
\mathbb{E}[(\mathbf{y}^\top \mathbf{a}_i \mathbf{a}_i^\top \mathbf{y})^2] = \frac{3}{d^2}\|\mathbf{y}\|^4.
$$

Noting $\mathbb{E}(\mathbf{y}^\top \mathbf{a}_i \mathbf{a}_i^\top \mathbf{y}) = d^{-1}\|\mathbf{y}\|^2$, we then have

$$
\mathbb{E}\left[d^2/D^2(\mathbf{y}^\top \mathbf{A}\mathbf{A}^\top \mathbf{y})^2\right] = \|\mathbf{y}\|^4 + \frac{2}{d}\|\mathbf{y}\|^4.
$$

(7) is from Lemma 3 □

**Proposition 2.** *For the Hashing Random Matrix, we rewrite $\mathbf{A} = \mathbf{S}\mathbf{D}$, where $\mathbf{S} \in \mathbb{R}^{d \times D}$ has each column chosen independently and uniformly from the $r$ standard basis vectors of $\mathbb{R}^d$ and $\mathbf{D} \in \mathbb{R}^{D \times D}$ is a diagonal matrix with diagonal entries chosen independently and uniformly on $b_i \in \{\pm 1\}$, then we have*

$$
\mathbb{E}\left[(\mathbf{x}^\top \mathbf{A}^\top \mathbf{A}\mathbf{x} - \mathbf{x}^\top \mathbf{x})^2\right] = d^{-1}\left(\|\mathbf{x}\|^4 - \sum_{i=1}^{D} x_i^4\right); \tag{10}
$$

$$
\mathbb{E}\left[(d/D\mathbf{y}^\top \mathbf{A}\mathbf{A}^\top \mathbf{y} - \mathbf{y}^\top \mathbf{y})^2\right] = D^{-1}\left(d\sum_{i=1}^{D} y_i^4 - d\|\mathbf{y}\|^4\right). \tag{11}
$$

*Proof.* Let $\mathbf{S} = (\boldsymbol{s}_1, \cdots, \boldsymbol{s}_D)$, such that $\boldsymbol{s}_i$ is a random vector taking the vector $e_j$ for equal probability, where $e_k$ is the $k$th standard unit vector of $\mathbb{R}^d$ for $k = 1, \cdots, d$. Then $\mathbb{E}(\boldsymbol{s}_i) = d^{-1}\mathbf{1}_d$ and $\mathbb{E}(\boldsymbol{s}_i^\top \boldsymbol{s}_i) = 1$, which follow that $\mathbb{E}[(\mathbf{S}^\top \mathbf{S})_{ij}] = \mathbb{E}(\boldsymbol{s}_i^\top \boldsymbol{s}_j) = \mathbb{E}(\boldsymbol{s}_i)^\top \mathbb{E}(\boldsymbol{s}_j) = d^{-2}$ for $i \neq j$; and $\mathbb{E}[(\mathbf{S}^\top \mathbf{S})_{ii}] = \mathbb{E}(\boldsymbol{s}_i^\top \boldsymbol{s}_i) = 1$.

Applying Lemma 2,

$$
\mathbb{E}\left[(\mathbf{x}^\top \mathbf{A}^\top \mathbf{A}\mathbf{x})^2\right] = (\mathbf{x}^\top \mathbf{x})^2 + d^{-1}\left(\|\mathbf{x}\|^4 - \sum_{i=1}^{D} x_i^4\right).
$$

Thus, (10) is proved.

Applying Lemma 3,

$$
\mathbb{E}\left[(d/D\mathbf{y}^\top \mathbf{A}\mathbf{A}^\top \mathbf{y} - \mathbf{y}^\top \mathbf{y})^2\right] = (\mathbf{y}^\top \mathbf{y})^2 + D^{-1}\left(d\sum_{i=1}^{d} y_i^4 - \|\mathbf{y}\|^4\right).
$$

Thus, (11) is proved. □

## 6.3 Consistency of Gaussian Process Fit from the original space to the embedding space

Besides of the isometry in expectation, we also examine the concentration of $\tilde{\mathbf{x}}$ around the original point $\mathbf{x}$ in terms of the function $f$. We measure the concentration by the difference between $\mathbf{x}^\top \tilde{\mathbf{x}}$ round $\mathbf{x}^\top \mathbf{x}$, which presents how much $\tilde{\mathbf{x}} - \mathbf{x}$ projects onto $\mathbf{x}$. Assume a matrix $\mathbf{A}$ satisfying Definition 1, we have

$$
\mathbb{E}\left[(\mathbf{x}^\top \mathbf{A}^\top \mathbf{A}\mathbf{x} - \mathbf{x}^\top \mathbf{x})^2\right] \leq 2d^{-1}\|\mathbf{x}\|^4. \tag{12}
$$

Assume a matrix $\mathbf{A}$ satisfying Definition 2, we have

$$
\mathbb{E}\left[(\mathbf{x}^\top \mathbf{A}^\top \mathbf{A}\mathbf{x} - \mathbf{x}^\top \mathbf{x})^2\right] \leq d^{-1}\left(\|\mathbf{x}\|^4 - \sum_{i=1}^{D} x_i^4\right). \tag{13}
$$

The proofs of (12) & (13) are provided in Lemmas 1 and 2, respectively.

For simplifying, we focus on the squared exponential kernel

$$K_{SE}(\mathbf{x}_1, \mathbf{x}_2) = \theta_0 \exp\{-2^{-1} r^2(\mathbf{x}_1, \mathbf{x}_2)\},$$

where

$$r^2(\mathbf{x}_1, \mathbf{x}_2) = \sum_{j=1}^{D} (x_{1j} - x_{2j})^2 / \theta_j^2.$$

Since $\theta_j$ can be absorbed into $x_{1j} - x_{2j}$, without loss of generality, we simplify to consider

$$r^2(\mathbf{x}_1, \mathbf{x}_2) = (\mathbf{x}_1 - \mathbf{x}_2)^\top (\mathbf{x}_1 - \mathbf{x}_2).$$

In the embedding space, $\mathbf{y}_1 = \mathbf{A}\mathbf{x}_1$ and $\mathbf{y}_2 = \mathbf{A}\mathbf{x}_2$, then the corresponding kernel is given by

$$r^2(\mathbf{y}_1, \mathbf{y}_2) = (\mathbf{x}_1 - \mathbf{x}_2)^\top \mathbf{A}^\top \mathbf{A}(\mathbf{x}_1 - \mathbf{x}_2).$$

According to (12) and (13), we have

$$\mathbb{E}[(r^2(\mathbf{y}_1, \mathbf{y}_2) - r^2(\mathbf{x}_1, \mathbf{x}_2))^2] \leq \frac{2}{d}[r^2(\mathbf{x}_1, \mathbf{x}_2)]^2.$$

By applying the Markov inequality, we obtain that there exists $\epsilon > \sqrt{2}$ such that

$$P\left(|r^2(\tilde{\mathbf{x}}_1, \tilde{\mathbf{x}}_2) - r^2(\mathbf{x}_1, \mathbf{x}_2)| \leq \epsilon d^{-1/2} r^2(\mathbf{x}_1, \mathbf{x}_2)\right) \geq 1 - \frac{2}{d} \frac{[r^2(\mathbf{x}_1, \mathbf{x}_2)]^2}{[\epsilon d^{-1/2} r^2(\mathbf{x}_1, \mathbf{x}_2)]^2}$$

$$= 1 - \frac{2}{\epsilon^2}.$$

Therefore, $r^2(\mathbf{y}_1, \mathbf{y}_2) = (1 + O_p(d^{-1/2})) r^2(\mathbf{x}_1, \mathbf{x}_2)$. It follows

$$\kappa(\mathbf{A}\mathbf{x}_1, \mathbf{A}\mathbf{x}_2) = (1 + O_p(d^{-1/2})) \kappa(\mathbf{x}_1, \mathbf{x}_2).$$

### 6.4 Consistency of Gaussian Process Fit from the embedding space to the original space

Similar to Appendix 6.3, we consider

$$r^2(\mathbf{y}_1, \mathbf{y}_2) = (\mathbf{y}_1 - \mathbf{y}_2)^\top (\mathbf{y}_1 - \mathbf{y}_2).$$

From the embedding space to the original space, $\tilde{\mathbf{x}}_1 = \mathbf{A}^\top \mathbf{y}_1$ and $\tilde{\mathbf{x}}_2 = \mathbf{A}^\top \mathbf{y}_2$. The corresponding kernel is then given by

$$r^2(\tilde{\mathbf{x}}_1, \tilde{\mathbf{x}}_2) = d^2/D^2 (\tilde{\mathbf{x}}_1 - \tilde{\mathbf{x}}_2)^\top (\tilde{\mathbf{x}}_1 - \tilde{\mathbf{x}}_2) = d^2/D^2 (\mathbf{y}_1 - \mathbf{y}_2)^\top \mathbf{A}\mathbf{A}^\top (\mathbf{y}_1 - \mathbf{y}_2).$$

Assume a matrix $\mathbf{A}$ satisfying Definition 1, (7) in Proposition 1 follows

$$\mathbb{E}\left[(d/D(\mathbf{y}_1 - \mathbf{y}_2)^\top \mathbf{A}\mathbf{A}^\top (\mathbf{y}_1 - \mathbf{y}_2) - \|\mathbf{y}_1 - \mathbf{y}_2\|^2)^2\right] \leq \frac{2}{D} \|\mathbf{y}_1 - \mathbf{y}_2\|^4. \tag{14}$$

According to (14), we have We have

$$\mathbb{E}[(r^2(\tilde{\mathbf{x}}_1, \tilde{\mathbf{x}}_2) - r^2(\mathbf{y}_1, \mathbf{y}_2))^2] \leq \frac{2}{D}[r^2(\mathbf{y}_1, \mathbf{y}_2)]^2.$$

By applying the Markov inequality, we obtain that there exists $\epsilon > \sqrt{2}$ such that

$$P\left(|r^2(\tilde{\mathbf{x}}_1, \tilde{\mathbf{x}}_2) - r^2(\mathbf{y}_1, \mathbf{y}_2)| \leq \epsilon D^{-1/2} r^2(\mathbf{y}_1, \mathbf{y}_2)\right) \geq 1 - \frac{2}{D} \frac{[r^2(\mathbf{y}_1, \mathbf{y}_2)]^2}{[\epsilon D^{-1/2} r^2(\mathbf{y}_1, \mathbf{y}_2)]^2}$$

$$= 1 - \frac{2}{\epsilon^2}.$$

Therefore, $r^2(\tilde{\mathbf{x}}_1, \tilde{\mathbf{x}}_2) = (1 + O_p(D^{-1/2}))r^2(\mathbf{y}_1, \mathbf{y}_2)$. It follows

$$\kappa(\mathbf{A}^\top \mathbf{y}_1, \mathbf{A}^\top \mathbf{y}_2) = (1 + O_p(D^{-1/2}))\kappa(\mathbf{y}_1, \mathbf{y}_2).$$

Similarly, assume a matrix $\mathbf{A}$ satisfying Definition 2, (11) in Proposition 2 follows

$$\mathbb{E}\left[(d/D(\mathbf{y}_1 - \mathbf{y}_2)^\top \mathbf{A}\mathbf{A}^\top(\mathbf{y}_1 - \mathbf{y}_2) - \|\mathbf{y}_1 - \mathbf{y}_2\|^2)^2\right] \leq D^{-1}\left(d\sum_{i=1}^d (y_{i1} - y_{i2})^4 - \|\mathbf{y}_1 - \mathbf{y}_2\|^4\right)$$
$$\leq \frac{d-1}{D}\|\mathbf{y}_1 - \mathbf{y}_2\|^4. \tag{15}$$

It follows

$$\kappa(\mathbf{A}^\top \mathbf{y}_1, \mathbf{A}^\top \mathbf{y}_2) = (1 + O_p((D/d)^{-1/2}))\kappa(\mathbf{y}_1, \mathbf{y}_2).$$

### 6.4.1 Three Lemmas

**Lemma 1.** *Let $\mathbf{A} = (\boldsymbol{l}_1, \cdots, \boldsymbol{l}_p)^\top$, such that $\boldsymbol{l}_i = (\alpha_{i1}, \cdots, \alpha_{iq})^\top$ is an $q \times 1$ vector where each element is from zero mean distribution with $\mathbb{E}(\alpha_{ij}^2) = 1$ and $\mathbb{E}(\alpha_{ij}^4) = \gamma$, we have that, for any vectors $\mathbf{x}_1 \in \mathbb{R}^q$ and $\mathbf{x}_2 \in \mathbb{R}^q$, where $x_{1i}$ and $x_{2i}$ are their $i$-th element, respectively.*

$$\mathbb{E}\left[(\mathbf{x}_1^\top \boldsymbol{l}_i \boldsymbol{l}_i^\top \mathbf{x}_2)^2\right] = \mathbf{x}_1^\top[(\gamma - 3)\mathbf{W}_2 + 2\mathbf{x}_2\mathbf{x}_2^\top + \|\mathbf{x}_2\|^2\mathbf{I}]\mathbf{x}_1, \tag{16}$$

*where $\mathbf{W}_2 = diag\{x_{21}x_{21}, \cdots, x_{2q}x_{2q}\}$. Particularly, for Gaussian projection,*

$$\mathbb{E}\left[(\mathbf{x}_1^\top \boldsymbol{l}_i \boldsymbol{l}_i^\top \mathbf{x}_2)^2\right] = 2\mathbf{x}_1^\top \mathbf{x}_2 \mathbf{x}_2^\top \mathbf{x}_1 + \|\mathbf{x}_2\|^2\|\mathbf{x}_1\|^2, \tag{17}$$

*Proof.* Since $\mathbf{x}_1^\top \boldsymbol{l}_i = \sum_{j=1}^q \alpha_{ij}x_{1j}$ and $\mathbf{x}_2^\top \boldsymbol{l}_i = \sum_{j=1}^q \alpha_{ij}x_{2j}$, we have

$$\mathbf{x}_1^\top \boldsymbol{l}_i \boldsymbol{l}_i^\top \mathbf{x}_2 = (\sum_{j=1}^q \alpha_{ij}x_{1j})(\sum_{j=1}^q \alpha_{ij}x_{2j}) = \sum_{j=1}^q \alpha_{ij}^2 x_{1j}x_{2j} + \sum_{j_1 \neq j_2} \alpha_{ij_1}\alpha_{ij_2}x_{1j_1}x_{2j_2}.$$

Noting $\mathbb{E}(\alpha_{ij}^4) = \gamma$, we have

$$\mathbb{E}\left[\left(\sum_{j=1}^q \alpha_{ij}^2 x_{1j}x_{2j}\right)^2\right] = \gamma \sum_{j=1}^q x_{1j}^2 x_{2j}^2 + \left(\sum_{j_1 \neq j_2} x_{1j_1}x_{2j_1}x_{2j_2}x_{1j_2}\right)$$
$$= \left(\sum_{j=1}^q x_{1j}^2\right)\left(\sum_{j=1}^q x_{2j}^2\right) + (\gamma - 1)\sum_{j=1}^q x_{1j}^2 x_{2j}^2.$$

We also have

$$\mathbb{E}\left(\sum_{j_1 \neq j_2} \alpha_{ij_1}\alpha_{ij_2}x_{1j_1}x_{2j_2}\right)\left(\sum_{j_1 \neq j_2} \alpha_{ij_1}\alpha_{ij_2}x_{2j_1}x_{1j_2}\right)$$
$$= \left(\sum_{j_1 \neq j_2} x_{1j_1}x_{2j_2}x_{2j_1}x_{1j_2}\right) + \sum_{j_1 \neq j_2} x_{1j_1}^2 x_{2j_2}^2$$
$$= \left(\sum_{j=1}^q x_{1j}x_{2j}\right)^2 - \sum_{j=1}^q x_{1j}^2 x_{2j}^2 + \left(\sum_{j=1}^q x_{1j}^2\right)\left(\sum_{j=1}^q x_{2j}^2\right) - \sum_{j=1}^q x_{1j}^2 x_{2j}^2;$$
$$\mathbb{E}\left(\sum_{j=1}^q \alpha_{ij}^2 x_{1j}x_{2j}\right)\left(\sum_{j_1 \neq j_2} \alpha_{ij_1}\alpha_{ij_2}x_{2j_1}x_{1j_2}\right) = 0;$$
$$\mathbb{E}\left(\sum_{j_1 \neq j_2} \alpha_{ij_1}\alpha_{ij_2}x_{1j_1}x_{2j_2}\right)\left(\sum_{j=1}^q \alpha_{ij}^2 x_{2j}x_{1j}\right) = 0.$$

Combing the four equations above, it is easy to verify that,

$$\mathbb{E}\left[(\mathbf{x}_1^\top \boldsymbol{l}_i \boldsymbol{l}_i^\top \mathbf{x}_2)^2\right] = (\gamma - 3)\sum_{j=1}^q x_{1j}^2 x_{2j}^2 + 2(\mathbf{x}_1^\top \mathbf{x}_2)^2 + \|\mathbf{x}_2\|^2\|\mathbf{x}_1\|^2.$$

$\square$

**Lemma 2.** *Consider the Hashing random projection* $\mathbf{A} = \mathbf{SD}$*, where* $\mathbf{S} \in \mathbb{R}^{d \times D}$ *has each column chosen independently and uniformly from the* $r$ *standard basis vectors of* $\mathbb{R}^d$ *and* $\mathbf{D} \in \mathbb{R}^{D \times D}$ *is a diagonal matrix with diagonal entries chosen independently and uniformly on* $b_i \in \{\pm 1\}$*. Then we have that for any vectors* $\mathbf{x}_1 \in \mathbb{R}^D$ *and* $\mathbf{x}_2 \in \mathbb{R}^D$*, where* $x_{1i}$ *and* $x_{2i}$ *are their* $i$*-th elements, respectively,*

$$\mathbb{E}\left[(\mathbf{x}_1^\top \mathbf{A}^\top \mathbf{A} \mathbf{x}_2)^2\right] = \mathbf{x}_1^\top \mathbf{x}_2 \mathbf{x}_2^\top \mathbf{x}_1 + d^{-1}\|\mathbf{x}_1\|^2\|\mathbf{x}_2\|^2 - d^{-1}\mathbf{x}_1^\top \mathbf{W}_2 \mathbf{x}_1.$$

*Proof.* Let $\mathbf{S} = (\boldsymbol{s}_1, \cdots, \boldsymbol{s}_D)$, such that $\boldsymbol{s}_i$ is a random vector taking the vector $\mathbf{e}_j$ for equal probability, where $\mathbf{e}_j$ is the $j$th standard unit vector of $\mathbb{R}^d$ for $j = 1, \cdots, d$. Then $\mathbb{E}(\boldsymbol{s}_i) = d^{-1}\mathbf{1}_d$ and $\mathbb{E}(\boldsymbol{s}_i^\top \boldsymbol{s}_i) = 1$, which follow that $\mathbb{E}[(\mathbf{S}^\top \mathbf{S})_{ij}] = \mathbb{E}(\boldsymbol{s}_i^\top \boldsymbol{s}_j) = \mathbb{E}(\boldsymbol{s}_i)^\top \mathbb{E}(\boldsymbol{s}_j) = \frac{1}{d^2}$ for $i \neq j$; and $\mathbb{E}[(\mathbf{S}^\top \mathbf{S})_{ii}] = \mathbb{E}(\boldsymbol{s}_i^\top \boldsymbol{s}_i) = 1$. We have that,

$$\mathbf{x}_1^\top \mathbf{A}^\top \mathbf{A} \mathbf{x}_2 = (\sum\nolimits_{i=1}^D b_i \boldsymbol{s}_i x_{1i})^\top (\sum\nolimits_{i=1}^D b_i \boldsymbol{s}_i x_{2i}) = \sum\nolimits_{i=1}^D b_i^2 x_{1i} \boldsymbol{s}_i^\top \boldsymbol{s}_i x_{2i} + \sum\nolimits_{i \neq j} b_i b_j x_{1i} \boldsymbol{s}_i^\top \boldsymbol{s}_j x_{2j} \quad (18)$$

From (18), we have,

$$\mathbb{E}\left[(\mathbf{x}_1^\top \mathbf{A}^\top \mathbf{A} \mathbf{x}_2)^2\right] = \mathbb{E}\left[(\sum\nolimits_{i=1}^D b_i^2 x_{1i} \boldsymbol{s}_i^\top \boldsymbol{s}_i x_{2i} + \sum\nolimits_{i \neq j} b_i b_j x_{1i} \boldsymbol{s}_i^\top \boldsymbol{s}_j x_{2j})^2\right]$$

$$= \sum\nolimits_{i=1}^D \mathbb{E}[(x_{1i} \boldsymbol{s}_i^\top \boldsymbol{s}_i x_{2i})^2] + \sum\nolimits_{i \neq j} \mathbb{E}(x_{1i} \boldsymbol{s}_i^\top \boldsymbol{s}_i x_{2i})(x_{1j} \boldsymbol{s}_j^\top \boldsymbol{s}_j x_{2j}) + \sum\nolimits_{i \neq j} \mathbb{E}[(x_{1i} \boldsymbol{s}_i^\top \boldsymbol{s}_j x_{2j})^2]$$

$$=: E_1 + E_2 + E_3.$$

Specifically, we have that

$$E_1 = \sum\nolimits_{i=1}^D \mathbb{E}[(x_{1i} \boldsymbol{s}_i^\top \boldsymbol{s}_i x_{2i})^2] = \sum\nolimits_{i=1}^D \sum\nolimits_{k=1}^d d^{-1}(x_{1i} \mathbf{e}_k^\top \mathbf{e}_k x_{2i})^2 = \sum\nolimits_{i=1}^D (x_{1i} x_{2i})(x_{1i} x_{2i})^\top.$$

$$E_2 = \sum\nolimits_{i \neq j} \mathbb{E}(x_{1i} \boldsymbol{s}_i^\top \boldsymbol{s}_i x_{2i}) \mathbb{E}(x_{1j} \boldsymbol{s}_j^\top \boldsymbol{s}_j x_{2j})^\top = \sum\nolimits_{i \neq j} (x_{1i} x_{2i})(x_{1j} x_{2j})$$

$$E_3 = \sum\nolimits_{i \neq j} \mathbb{E}[(x_{1i} \boldsymbol{s}_i^\top \boldsymbol{s}_j x_{2j})^2] = \sum\nolimits_{i \neq j} x_{1i}^2 x_{2j}^2 \mathbb{E}(\boldsymbol{s}_i^\top \boldsymbol{s}_j \boldsymbol{s}_j^T \boldsymbol{s}_i) = \sum\nolimits_{i \neq j} x_{1i}^2 x_{2j}^2 d^{-1} \mathbb{E}(\boldsymbol{s}_i^\top \boldsymbol{s}_i)$$

$$= d^{-1} \sum\nolimits_{i \neq j} x_{1i}^2 x_{2j}^2$$

$$= d^{-1}\|\mathbf{x}_1\|^2\|\mathbf{x}_2\|^2 - d^{-1} \sum\nolimits_{i=1}^D x_{1i}^2 x_{2i}^2. \quad (19)$$

Thus, we have,

$$\mathbb{E}\left[(\mathbf{x}_1^\top \mathbf{A}^\top \mathbf{A} \mathbf{x}_2)^2\right] = \mathbf{x}_1^\top \mathbf{x}_2 x_{2i} \mathbf{x}_1 + d^{-1}\|\mathbf{x}_1\|^2\|\mathbf{x}_2\|^2 - d^{-1} \sum\nolimits_{i=1}^D x_{1i}^2 x_{2i}^2.$$

$\square$

**Lemma 3.** *Consider the Hashing random projection* $\mathbf{A} = \mathbf{SD}$*, where* $\mathbf{S} \in \mathbb{R}^{d \times D}$ *has each column chosen independently and uniformly from the* $r$ *standard basis vectors of* $\mathbb{R}^d$ *and* $\mathbf{D} \in \mathbb{R}^{D \times D}$ *is a diagonal matrix with diagonal entries chosen independently and uniformly on* $b_i \in \{\pm 1\}$*. Then we have that for any vectors* $\mathbf{y}_1 \in \mathbb{R}^d$ *and* $\mathbf{y}_2 \in \mathbb{R}^d$*,*

$$\mathbb{E}\left[D^{-2} d^2 (\mathbf{y}_1^\top \mathbf{A} \mathbf{A}^\top \mathbf{y}_2)^2\right] = (\mathbf{y}_1^\top \mathbf{y}_2)^2 + D^{-1}\left[d \sum\nolimits_{k=1}^d (y_{1k} y_{2k})^2 - (\mathbf{y}_1^\top \mathbf{y}_2)^2\right].$$

*Proof.* Let $\mathbf{S} = (\boldsymbol{s}_1, \cdots, \boldsymbol{s}_D)$, such that $\boldsymbol{s}_i$ is a random vector taking the vector $\mathbf{e}_j$ for equal probability, where $\mathbf{e}_j$ is the $j$th standard unit vector of $\mathbb{R}^d$ for $j = 1, \cdots, d$. Then $\mathbb{E}(\boldsymbol{s}_i) = d^{-1}\mathbf{1}_d$ and $\mathbb{E}(\boldsymbol{s}_i \boldsymbol{s}_i^\top) = d^{-1}\mathbf{I}$. We have that,

$$\mathbf{y}_1^\top \mathbf{A} \mathbf{A}^\top \mathbf{y}_2 = \sum\nolimits_{i=1}^D b_i^2 \mathbf{y}_1^\top \boldsymbol{s}_i \boldsymbol{s}_i^\top \mathbf{y}_2. \quad (20)$$

From (20), we have,

$$\mathbb{E}\left[(\mathbf{y}_1^\top \mathbf{A}\mathbf{A}^\top \mathbf{y}_2)^2\right] = \mathbb{E}\left[\left(\sum_{i=1}^D b_i^2 \mathbf{y}_1^\top \boldsymbol{s}_i \boldsymbol{s}_i^\top \mathbf{y}_2\right)^2\right]$$

$$= \sum_{i=1}^D \mathbb{E}[(\mathbf{y}_1^\top \boldsymbol{s}_i \boldsymbol{s}_i^\top \mathbf{y}_2)^2] + \sum_{i\neq j} \mathbb{E}(\mathbf{y}_1^\top \boldsymbol{s}_i \boldsymbol{s}_i^\top \mathbf{y}_2)(\mathbf{y}_1^\top \boldsymbol{s}_j \boldsymbol{s}_j^\top \mathbf{y}_2)$$

$$=:E_1 + E_2.$$

Specifically, we have that

$$E_1 = \sum_{i=1}^D \mathbb{E}[(\mathbf{y}_1^\top \boldsymbol{s}_i \boldsymbol{s}_i^\top \mathbf{y}_2)^2] = D\sum_{k=1}^d d^{-1}(\mathbf{y}_1^\top \mathbf{e}_k \mathbf{e}_k^\top \mathbf{y}_2)^2 = d^{-1}D\sum_{k=1}^d (y_{1k}y_{2k})^2.$$

$$E_2 = \sum_{i\neq j} \mathbb{E}(\mathbf{y}_1^\top \boldsymbol{s}_i \boldsymbol{s}_i^\top \mathbf{y}_2)\mathbb{E}(\mathbf{y}_1^\top \boldsymbol{s}_j \boldsymbol{s}_j^\top \mathbf{y}_2) = d^{-2}(D^2 - D)(\mathbf{y}_1 \mathbf{y}_2^\top)^2. \tag{21}$$

Thus, we have,

$$\mathbb{E}\left[D^{-2}d^2(\mathbf{y}_1^\top \mathbf{A}\mathbf{A}^\top \mathbf{y}_2)^2\right] = (\mathbf{y}_1^\top \mathbf{y}_2)^2 + D^{-1}\left[d\sum_{k=1}^d (y_{1k}y_{2k})^2 - (\mathbf{y}_1^\top \mathbf{y}_2)^2\right].$$

$\square$

## 6.5  Details of machine information

The entire experiment in this paper is programmed in Python and run under the Linux system, using open-source Bayesian optimization libraries such as Ax(Bakshy et al., 2018) and BoTorch(Balandat et al., 2020) for assistance. The experimental equipment is equipped with 2 AMD EPYC 7601 processors, each with 32 cores and a base clock frequency of 2.2GHz. The experiment is conducted in the form of parallel processing, with different independent repeating experiments distributed to different CPU cores for acceleration. In addition, the system has a memory capacity of 768GB, providing sufficient memory space for large-scale data processing and complex algorithm operation.

## 6.6  Modified Schwefel function and Griewank function

To investigate the efficacy of the proposed method within the complex context of high-dimensional optimization problems that entail numerous local minima, we applied a Schwefel function of $D = 100$ and a Griewank function of $D = 100$. In the context of the Schwefel function and Griewank function, every dimension qualifies as an effective dimension, hence $d_e = D = 100$. We increased the optimization challenge by altering the Schwefel function and Griewank function. This alteration involved the adjustment of positions within different dimensions where minimum values are attained, thereby making the optimization of the Schwefel function and Griewank function more challenging. The modified Schwefel function is:

$$f(\mathbf{x}) = \sum_{i=1}^D \frac{x_i^2}{4000} - \prod_{i=1}^d \cos\left(\frac{x_i}{\sqrt{i}}\right) + 1, \tag{22}$$

Where $b_i \sim \mathcal{N}(0,1)$. We maintain the constancy of $b_i$ values across different independent repeated experiments, while allowing $b_i$ values to vary across different dimensions.

The modified Griewank function can be expressed as:

$$f(\mathbf{x}) = \sum_{i=1}^D w_i(x_i - b_i)^2 - \prod_{i=1}^d \cos\left(\frac{x_i}{\sqrt{i}}\right), \tag{23}$$

where $w_i \sim \mathcal{N}(0,1)$ and $b_i \sim \mathcal{N}(0,1)$. We ensure that the values of $w_i$ and $b_i$ remain consistent within various independent repeat experiments, while differing across the several dimensions.

### 6.7 Details about Real-World Problems

#### 6.7.1 Lunar landing

In this experiment, our goal is to learn a strategy that controls the lunar lander, so that the lunar lander can minimize fuel consumption and distance from the landing target, while avoiding crashes. This optimization task was proposed by Eriksson(Eriksson et al., 2019). The simulation environment of the control task is implemented through OpenAI gym [1]. The state space of the lunar lander includes its coordinates $x$ and $y$, linear velocities $x_v$ and $y_v$, its angle, its angular velocity, and two boolean values indicating whether each leg is in contact with the ground. At any moment, the current controller state can be represented with an 8-dimensional vector. After obtaining the state vector, the controller can choose one of four possible actions, corresponding to pushing the thrusters left, right, up or none. In the experiment, it can be considered as a $D = 12$ optimization problem. Once the parameters are determined, the corresponding rewards can be obtained through in-game feedback. If the lander deviates from the landing pad, it loses rewards. If the lander crashes, it gets an extra $-100$ points. If it successfully controls the lander to stop, it will get an extra $+100$ points. Each leg touching the ground gets $+10$ points. Igniting the main engine gets $-0.3$ points per frame. Each frame starts side engine for $-0.03$ points. The goal of the control task optimization is to maximize the average final reward on a fixed set of 50 randomly generated terrains, initial positions, and speed combinations. We observe that even minor perturbations can have an impact on the simulation.

#### 6.7.2 Robot pushing

This paper follows the experimental setup of Wang(Wang et al., 2017), Eriksson(Eriksson et al., 2019) et al., and also realizes the simulation of using two robot arms to push two objects in the Box 2D(Catto, 2011) physics engine. In the simulation environment, the parameters of the robot arms are simulated to push two objects, and the trajectories of the object movements are recorded at the same time. A total of 14 parameters are used by the two robot arms, which respectively specify the position and rotation of the robot hands, the pushing speed, the moving direction, and the pushing time. The lower bounds for these parameters are

$$[-5, -5, -10, -10, 2, 0, -5, -5, -10, -10, 2, 0, -5, -5],$$

and the upper bounds are

$$[5, 5, 10, 10, 30, 2\pi, 5, 5, 10, 10, 30, 2\pi, 5, 5].$$

The initial positions of the objects are designated as $s_{i0}$ and $s_{i1}$, and the end positions as $s_{e0}$ and $s_{e1}$. The target positions of the two objects are indicated by $s_{g0}$ and $s_{g1}$. The reward is defined as

$$r = |s_{g0} - s_{i0}| + |s_{g1} - s_{i1}| - |s_{g0} - s_{e0}| - |s_{g1} - s_{e1}|,$$

namely the distance by which the objects move towards their target positions.

#### 6.7.3 NAS

In this paper, referring to the settings of Letham(Letham et al., 2020) and others, by parameterizing operations and edges respectively, the optimal architecture search problem in NASBench-101 is set as a continuous high-dimensional Bayesian optimization problem. Specifically, $L$ different operations are represented by one-hot encoding.

Since two of the seven nodes are fixed as input and output nodes, the remaining five optional nodes, each node corresponds to three different operations, which generate a total of 15 different parameters. We optimize these parameters in the continuous $[0, 1]$ space. For each node, we take the "operation" corresponding to the maximum value of the three operations under that node as the "operation" adopted by that node, and use one-hot encoding to represent the specific "operation" used under that node.

Since NASBench-101 uses a $7 \times 7$ upper triangular adjacency matrix to represent edges, it generates a total of $\frac{7 \cdot 6}{2} = 21$ possible edges. And the five optional vertices can have three different operations, so under this

---

[1] www.gymlibrary.dev/environments/box2d/lunar_lander/

encoding there are about $2^{21} \cdot 3^5 \approx 510M$ unique models, After removing a large number of unreasonable input and output models and models with more than 9 edges, the search space still has about $423k$ unique models. We convert these 21 possible edges into 21 binary parameters that are similarly optimized in a continuous $[0, 1]$ space. We rank the continuous values corresponding to these 21 binary parameters and create an empty adjacency matrix. Then, we add edges to the adjacency matrix in the percentile order of the 21 binary parameters iteratively, while pruning parts that are not connected to the input or output nodes, until reaching the limit of 9 edges. Finally, the combination of adjacency matrix parameters (21) and one-hot encoded "operation" parameters (15) constitutes a 36-dimensional optimization space. The Bayesian optimization algorithm only needs to be optimized in a high-dimensional space with $D = 36$, and the boundary constraint is $[-1, 1]^{36}$. Each vector $\mathbf{x} \in \mathbb{R}^{36}$ can be decoded into a DAG and lookup evaluated in NASBench-101.

### 6.7.4   Rover planning

To explore the performance of the proposed method in complex high-dimensional optimization scenarios, we considered a two-dimensional trajectory optimization task aimed at simulating detector navigation missions. This optimization task was proposed by Wang(Wang et al., 2018), and the experimental setup by Wang(Wang et al., 2018) was continued to be used here, with the optimization objective being to maximize the reward function. The problem instance is described by defining the starting position $s$, the target position $g$, and a cost function on the state space. The goal of the problem is to optimize the detector's trajectory on rugged terrain. The trajectory consists of a set of points on a two-dimensional plane, and there are 30 points in this instance, which can be fitted into a B-spline curve, so it is considered a high-dimensional optimization problem with $D = 60$. Through a set of trajectories, $\boldsymbol{x} \in [0, 1]^{60}$, and a specific cost function, we can calculate the cost of a trajectory $c(\boldsymbol{x})$.

The reward for this problem is defined as

$$f(\boldsymbol{x}) = c(\boldsymbol{x}) + \lambda \left( |\boldsymbol{x}_{0,1} - s| \, 1 + |\boldsymbol{x}_{59,60} - g|_1 \right) + b.$$

The reward function is non-smooth, discontinuous, and concave. The four input dimensions involved in the reward function respectively represent the starting and target positions of the trajectory. Set $\lambda = -10, b = 5$, any collision with objects along the trajectory will incur a penalty of $-20$, which is the collision cost of the trajectory. Thus, in addition to penalties in the reward function caused by collisions, adverse deviations from the trajectory's starting point will also incur additional penalties.

