# OpenReview forum: "High-dimensional Bayesian Optimization via Condensing-Expansion Projection"
_TMLR — Rejected by TMLR_

### Review · Reviewer_mE4T · 2025-01-27

**Summary Of Contributions:**

A recurrent challenge in global optimization is to handle many variables. One option to address the issue is to rely on random linear projections. In the Bayesian optimization (BO) community, this idea has been proposed by Wang et al., 2016 with the REMBO approach. Under the assumption that the function is intrinsically of low dimensionality, the main idea of REMBO is to rely on a few fixed random embeddings to conduct BO. Here the proposition is to use a different random embedding at each iteration. The authors compare their variant to the original methods on a few test problems.

**Audience:**

Yes

**Claims And Evidence:**

No

**Requested Changes:**

All these points are critical.

 Why calling “Condense-Expansion Projection” the standard linear projection with matrices A or A^T ? It is used in the Wang et al. 2016 REMBO paper, say in Figure 3 and Algorithm 2. In addition, the original REMBO paper is Wang, Z., Zoghi, M., Hutter, F., Matheson, D., & De Freitas, N. (2013, August). Bayesian optimization in high dimensions via random embeddings. In Proceedings of the Twenty-Third international joint conference on Artificial Intelligence (pp. 1778-1784).

The related work section is missing important references:
- lineBO, a 1d version of what is proposed here, is Kirschner, J., Mutny, M., Hiller, N., Ischebeck, R., & Krause, A. (2019, May). Adaptive and safe Bayesian optimization in high dimensions via one-dimensional subspaces. In International Conference on Machine Learning (pp. 3429-3438). PMLR.
- Eriksson, D., & Jankowiak, M. (2021, December). High-dimensional Bayesian optimization with sparse axis-aligned subspaces. In Uncertainty in Artificial Intelligence (pp. 493-503). PMLR.
- Hvarfner, C., Hellsten, E. O., & Nardi, L. (2024). Vanilla Bayesian Optimization Performs Great in High Dimension. arXiv preprint arXiv:2402.02229. ICML 2024
This latter should be discussed and makes a necessary baseline for the empirical comparison.

P1: what is the link between the curse of dimensionality and the computational cost associated with calculating the acquisition function?

P2: “However, unlike these studies, our approach select the new point in the original space, implying that our approach does not relies on the effective space assumption.” → This is incorrect, the approach select points on a random embedding, step 8 of Algorithm 1.

P2: “the possibility that the global optimum may not be located in a single embedding space.” What does this mean?

Figure 1: what are the axis? The difference between the left and right subfigures is too tiny.

Eq. (3) and associated paragraph: as discussed for instance by Letham at al., 2020 this is an effect of the Johnson-Lindenstrauss lemma.

Section 3.3
- First, REMBO and following works suggests using several random embeddings. This can be linked to theoretical results from Cartis, C., Massart, E., & Otemissov, A. (2023). Bound-constrained global optimization of functions with low effective dimensionality using multiple random embeddings. Mathematical Programming, 198(1), 997-1058. Using several fixed embedding should be added to the comparison.
- Then a key aspect, not mentioned, is that the main advantage of keeping the random embedding fixed is that a deterministic Gaussian process model can be used (even the appropriate kernel as with ALEBO). This cannot be done here unless using the correct linear embedding, hence it requires adding a noise term to account for the effect of the remaining dimensions. So the result on the covariance is misleading and a proper discussion of the associated limitations should be added.
- Why using the term trajectory for the observed data?

Section 3.4
-The use of the convex projection is already present in the REMBO paper. Further discussions and remedies are introduced in Binois et al. 2020 and in Letham et al., 2020. Also there is no such problem for hashing matrices as detailed by Nayebi et al., 2020.
-In practice, the proposed scaling strategy reduces the search space to the center of the domain. This is not properly mentioned.

Section 4
- Random search is missing as a baseline, as is a standard BO method (e.g., from Hvarfner et al.), SAASBO, and lineBO. Then REMBO includes several variants with different covariance kernels and the use of several random embedding. These are missing here.
- Three synthetic functions are not sufficient to validate an optimization method. How are they chosen? Specifically, since the author focus on linear embedding approaches, these functions do not really fit. In particular, the Schwefel and Griewank functions are additive and their solution is at the center of the domain, favoring the simple scaling approach proposed. None of these test functions includes a non-trivial random embedding (i.e., not solely adding useless variables) while this should be tested. Especially since it is the focus of the approach.
- The regret to the true solution rather than the values should be represented. On real world problems, how does it compare to the best known solutions?
- The optimization budget of 50 is uncommonly small, what is the reason for this choice?
- Noisy reward: how does the noise variance relate to the scale of the function values?
- “This suggests that for optimization issues with moderate to low dimensionalities, escalating the dimensions of the embedding subspaces can notably bolster the algorithms’ efficacy.” This has already been observed in the cited references, and also by Cartis et al.

Section 5:
- “This paper proposes a Bayesian optimization framework utilizing the Condensing-Expansion Projection technique, free from reliance on the assumption of effective dimension.” Do you have a theoretical convergence result to back this statement?
- “This flexibility enables the selection of a suitable projection matrix according to the problem’s characteristics beforehand.” What is it referring to in the paper?

Typos:
Abstract: does not reply → rely
P1: next point to evalution
p2: arg max is a set
p3: orginal; Figure1 (space)
p4: we also concern the concentration; sucha + other problems on the same line
Letham et al., 2020a and 2020b are the same paper
Figure 3: the text cannot be read

**Strengths And Weaknesses:**

Strengths:
- High dimension is a topic of interest for many practical applications.

Weaknesses:
- The paper is hard to read due to many imprecise statements.
- The state of the art is incomplete.
- The novelty with respect to the state of the art is incremental, and some existing results are not clearly stated as such.
- The the empirical comparison is too limited.

All these points are detailed below in the requested changes.

---

> ### Author Response · Authors · 2025-03-15
>
> Thank you for numerous helpful comments to enhance our manuscript. We appreciate them.
>
> - On the name of “Condense-Expansion Projection”
>
> Yes, our approach uses two standard random linear projections. We call it “Condense-Expansion Projection” as a figurative expression to capture the process of first reducing the dimensionality and then restoring it. We have clearly emphasized this point in the first paragraph of Section 3.2. We are also open to changing the name if necessary.
>
> - On the references
>
> Thank you for providing these references. We have incorporated them in Section 2. Additionally, we have modified or removed some of the strong statements you pointed out and highlighted the changes. Please review them.
>
> - On iteratively generating a new projection matrix at each iteration
>
> Thank you for your perspective and the additional information for comparison. We have added clarification on the motivation in the second paragraph of Section 3.3.
>
> - On the use of the convex projection in Section 3.4
>
> We have referenced these papers at the beginning of the second paragraph. Thank you for pointing this out.
>
> - On the analysis and the claim that the assumption of effective dimension
>
> In the previous version, we only provided the concentration of the transformed $\tilde{x}$ to the original $x$, which was indeed insufficient. In this version, we have presented an analysis of the approximate error of the GP fit to show that the CEP approach preserves the variance function of a Gaussian process. The new analysis also explains why CEP approach does not rely on the effective subspace assumption.
>
> We believe this revision addresses your comments. Please refer to the highlighted part in Section 3.2. Thank you for your insightful feedback, which has greatly motivated our improvements.
>
> - On Figure 1
>
> The confusion arises from the lack of description of the function values in the figure. In these plots, the colors represent function values. We have re-plotted the figure for better clarity.
>
> - On experiments
>
> In addition to the subspace-based algorithms, we have included two recent advanced algorithms operating in the original space, SAASBO and VanillaBO. We have reported their performance and provided comparisons with our approaches. Thank you for your suggestions.
>
> For the choice of synthetic functions, we follow some standard benchmarks. Although the functions in the simulation are limited, we also validate the performance on real datasets where the functions may be more complex. Due to time constraints, we did not consider additional synthetic functions in this revision, but we are open to incorporating more complex synthetic functions later if necessary.
>
> Following the approach in ALEBO (Letham et al., 2020), we typically chose a budget of 50 in the simulation studies. This budget is sufficient to observe the comparison. We will increase the budget if needed.
>
> For the Noisy reward, the signal-to-noise is about 9.
>
> We have also made other modifications.
>
> - On Section 5
>
> We have removed the strong statements as suggested.

---

> > ### Comment · Reviewer_mE4T · 2025-03-17
> >
> > Thank you for your detailed reply to my comments and adding new results. Some issues have been addressed but the main ones remain.
> >
> > -  On the name of “Condense-Expansion Projection”
> > > Yes, our approach uses two standard random linear projections. We call it “Condense-Expansion Projection” as a figurative expression to capture the process of first reducing the dimensionality and then restoring it. We have clearly emphasized this point in the first paragraph of Section 3.2. We are also open to changing the name if necessary.
> >
> > I don’t understand what is the difference with REMBO: “However, unlike these studies, our approach selects the new point in the embedding space and projects it back to the original space to obtain a point in the original space.” This is already done in REMBO, design points are in the high-dimensional space. They are projected in the low dimensional space within the covariance.
> >
> > - Figure 1: there is still no real difference between left and right, except that the violet point moved. Perhaps show it first in 1d.
> >
> > - My comment on using several random embeddings for the default REMBO method has not been addressed. The interest of keeping embeddings fixed to avoid estimating a noise term is still not discussed.
> >
> > - "In addition to the subspace-based algorithms, we have included two recent advanced algorithms operating in the original space, SAASBO and VanillaBO. We have reported their performance and provided comparisons with our approaches. Thank you for your suggestions."
> >
> > Thank you for these additions. It appears that vanillaBO is better than the proposed algorithm in many instances. The under-performance in some cases is probably due to the low number of evaluations used. Also there is no real reason to discard it in dimension 1000 (there are such results in the Hvarfner et al.,2024 paper.)
> >
> > - > For the choice of synthetic functions, we follow some standard benchmarks. Although the functions in the simulation are limited, we also validate the performance on real datasets where the functions may be more complex. Due to time constraints, we did not consider additional synthetic functions in this revision, but we are open to incorporating more complex synthetic functions later if necessary.
> >
> > It is necessary given the mitigated results when adding regular BO. My previous comment still holds. Same for plotting the regret.
> >
> > - > Following the approach in ALEBO (Letham et al., 2020), we typically chose a budget of 50 in the simulation studies. This budget is sufficient to observe the comparison. We will increase the budget if needed.
> >
> > Please do. The budget are higher for the more realistic test cases anyway.
> >
> >
> > - Section 3.4: how does it perform if the solution is on the boundary of the domain? Can you specify what is the domain \mathcal{Y} in practice? From reading the paragraph it seems that the rescaling is used to avoid the convex projection, but this is detrimental if the solution is on the boundary.
> >
> > - There is a contradiction between: P2 “However, GPs are known to predict poorly for large dimension D (Wang et al., 2016), which prevents the use of standard BO in high dimensions” and P5 “Hvarfner et al. (2024) demonstrated that appropriately scaling the lengthscale prior of the GP kernel makes vanilla BO perform well in high dimensions.”
> >
> > - > “as fitting the GP in such high=dimensional setting is costly and leads to significantly longer computation time.”
> >
> > With an isotropic covariance kernel this is not the case.
> >
> > - New typos:
> >
> > P2: sub- problem do- main contain
> >
> > P7: Comparative the performance
> >
> > P8: high=dimensional setting

---

> ### Author Response · Authors · 2025-04-01
>
> Thank you so much for your additional comments, and apologies for the delayed response. I had assumed we couldn’t provide further replies.
>
> - on the difference with REMBO
>
> Yes, in REMBO, the design points are in the high-dimensional space and are projected into a lower-dimensional space based on the covariance. This allows them to evaluate the objective function in the low-dimensional space under the assumption of an effective subspace. The key difference in our approach is that we use two projections to avoid relying on the effective subspace assumption.
>
> - on the budget and others
>
> We will provide the results by doubling the number of evaluations in the revised version. We expect the results to be consistent. For the case with a dimension of 1000, we did not include the results due to time limitations. In the further revision, we will address these experimental adjustments.

---

> > ### Comment · Reviewer_mE4T · 2025-04-01
> >
> > * on the difference with REMBO
> > First there are two variants of covariance kernels in REMBO. Please provide a clearer answer. REMBO usually starts from scratch but the use of existing data is transparent.
> > Then at every step the proposed approach relies on the effective subspace assumption.

---

> > > ### Author Response · Authors · 2025-04-02
> > >
> > > - on the comparison with the high-dimensional kernel of REMBO
> > >
> > > Yes, REMBO has two variants of kernels. My previous response referred to the low-dimensional kernel.
> > > For its high-dimensional kernel, which is calculated in $\mathcal{X}$, its search space is $\{p_{\mathcal{X}}(Ay): y\in\mathcal{Y}\}$, not $\mathcal{X}$, where $A\in\mathbb{R}^{D\times d}$ in their notation.
> > >
> > > In contrast, our approach calculates the kernel in $\mathcal{Y}$, and searches for points in $\mathcal{Y}$ too. And our approach obtains the function $f(x)$ values at $x=A^{\top}y$ over $\mathcal{X}$ by projecting back, using $A\in\mathbb{R}^{d\times D}$ in our notation.
> > >
> > > Let me know if anything needs further clarification.

---

> > > > ### Comment · Reviewer_mE4T · 2025-04-02
> > > >
> > > > REMBO also evaluates points in X, since it is necessary to evaluate the function. At a given iteration, everything is the same as in REMBO. The only change is that the embedding is not fixed . This impacts the quality of the model since the data is not on the search embedding. This is supposedly captured by the noise term but this effect is not discussed. For instance, is it better to use several embedding in parallel with exact GPs or to change it every time with noisy GPs?

---

> > > > > ### Author Response · Authors · 2025-04-03
> > > > >
> > > > > Thank you so much.
> > > > >
> > > > > I still don’t think our approach is the same as REMBO at a given iteration. There may be some misunderstandings or differences in viewpoints.
> > > > >
> > > > > Techinically, from the chosen $y^*$ in $\mathcal{Y}$ to $x^*$ in $\mathcal{X}$ for evaluating $f(x^*)$, we achieve this through the projection $A^{\top}$ on $y^*$.
> > > > >
> > > > > Regarding running multiple embeddings in parallel with exact GPs versus changing the embedding at each iteration with noisy GPs, our comparison with REMBO shows that the latter performs better. Although we did not explicitly analyze the sources of this outperformance, this is an interesting point worth exploring.

---

> ### Comment · Reviewer_mE4T · 2025-04-03
>
> > Techinically, from the chosen [...] on $y^*$.
>
> Like REMBO.
>
> > Regarding running multiple embeddings in parallel with exact GPs versus changing the embedding at each iteration with noisy GPs, our comparison with REMBO shows that the latter performs better.
>
> Unless I am missing something, your comparison is against REMBO with a single embedding.

---

> > ### Author Response · Authors · 2025-04-04
> >
> > Sorry for my misunderstanding. I double-checked it. Yes, in the implementation of REMBO, we chose a single matrix rather than multiple interleaved runs. In the revised version, we will include this comparison.

---

### Review · Reviewer_JcdP · 2025-01-29

**Summary Of Contributions:**

The paper proposes a random projection based high dimensional Bayesian optimization (BO). The surrogate model (Gaussian process) is estimated in the low dimensional space embedded by a random linear projection. The main claim is that the proposed method can avoid the effective subspace assumption, which is required by existing studies, by using different random projection at every iteration.

**Audience:**

Yes

**Claims And Evidence:**

No

**Requested Changes:**

- I currently do not think the error analysis of Section 3.2 justifies the proposed method. More convincing description why and how it justifies the proposed method should be required.
  - The benefit of the inequality bounds (2)-(3) are unclear. What does bounding $x^T \tilde{x} - x^T x$ imply about the proposed method? Can anything be stated about the more direct reconstruction error $\| \tilde{x} - x \|$ or the accuracy of the GP in the embedded space, or performance of final BO?
  - It is not fully clarified how small the right hand side of (2) and (3) is. In (2), at least up to d = 2, the inequality is trivial because it holds even if A = 0. On the other hand, if d = D, the reconstruction error can be reduced to zero by making A an invertible matrix and reconstructing using $A^{-1}$. However, the bounds (2)-(3) still have errors even when d = D (In the experiments, many of D is below 100, which seems to be a manageable size for considering the inverse). This also makes the smallness unclear. Further, d is typically assumed to be quite small (and d << D). The justification based on the condition that 'd is not small' is not convincing.
- The authors claim that the assumption of effective dimension can be removed. However, the rationale for dimensionality reduction without the effective dimension is not sufficiently revealed. A straightforward guess is that forcing dimensionality reduction without having an effective dimension would likely degrade the surrogate model accuracy, because usually f(x) is even not a function in the embedded space (multiple f exists for a single embedded point as shown in Fig.1) by which the assumption behind the GP is broken. A qualitative explanation why this degrade would not happen should be clarified. In other words, why is the embedding possible when the objective function does not have any low dimensional structure.

- In Section 3.3, the authors mention 'As demonstrated in Section 3.2, $\tilde{x}^*_{t}$ concentrates to $x^*_{t'}$. What does 'concentrates' mean? I currently do not think that Section 3.2 guarantees 'concentration' of $\tilde{x}$ around x, as already mentioned.

- Figure 1 is confusing. What is each axis of the plots. According to the caption D = 3 and d = 2. This means that the axis of 3D-original space is $x_1, x_2$, and $x_3$, and 2D-embedding space is $y_1$ and $y_2$? But, then, what is the surface? Or, are they D = 2 and d = 1?

- I guess in the center plot of Figure 1, the horizontal axis is the embedded space and the vertical axis is the objective function value. If so, I think this figure appears to overly emphasize only a rare case where the embedding surprisingly works well, making it seem far from what is actually expected to occur. It is unlikely that such an ideal situation would arise for most function shapes or projection directions. In this figure, the value that attains the minimum in the latent space happens to be restored as the minimum in the original space. However, it is unusual, because while the reconstruction error is bounded only in one direction (further, it may not small as already mentioned), there is no bound for the errors in other directions (I guess the minimum exists in the edge of the domain also causes this misleading illustration). If my guess is correct, this figure or its explanation should be revised to ensure scientific fairness.

- In experiments, the authors repeatedly mentioned that the original dimension of the benchmark function is 'extended' or 'expanded' to D dimension. Please clarify how was it extended/expanded.
- The vanilla BO without dimension reduction should be an informative baseline in experiments.

Minor issues:

- In abstract: 'does not reply' is 'does not rely'?
- After the period of the first sentence of Sec 2, white space is missed.
- In Sec2, the fourth line of the second paragraph: The name of the first author is not written: 'M. \& Krause'.
- The five points in the left and right plots of Figure 1 is difficult to see.
- Fonts in Figure 3 is too small.

**Strengths And Weaknesses:**

S: The paper is easy to follow, and well-organized.

S: The problem setting is quite important, and the proposed method is easy to implement.

W: The fundamental justification for the proposed method is unclear. The authors claim that changing the random projection at each iteration allows the removal of the effective dimension assumption. However, it is not clarified under what assumptions and why dimensionality reduction can be justified without assuming the effective dimension. Unless this point is clarified, the main claim (the proposed method does not require the effective dimension assumption unlike conventional methods) appears to be overstated.

W: An error analysis for the random projection is provided, but it seems incomplete. The discussion is limited to the reconstruction error in a specific projection direction that does not provide any guarantees for the proposed method. Claiming this analysis as a theoretical justification for the proposed method does not appear to be a scientifically valid argument.

---

> ### Author Response · Authors · 2025-03-15
>
> Thank you for numerous constructive comments to enhance our manuscript. .
>
> - On the analysis and the claim that the assumption of effective dimension
>
> In the previous version, we only provided the concentration of the transformed $\tilde{x}$ to the original $x$, which was indeed insufficient. In this version, we have presented an analysis of the approximate error of the GP fit to show that the CEP approach preserves the variance function of a Gaussian process. The new analysis also explains why CEP approach does not rely on the effective subspace assumption.
>
> We believe this revision addresses your comments. Please refer to the highlighted part in Section 3.2. Thank you for your insightful feedback, which has greatly motivated our improvements.
>
> - On Figure 1
>
> The confusion arises from the lack of description of the function values in the figure. In these plots, the colors represent function values. We have re-plotted the figure for better clarity.
>
> - On experiments
>
> The Holder Table function is a two-dimensional function, while BO attempts to fit it in a $D$-dimensional space.
> In contrast, the Schwefel and Griewank functions are defined over the entire $D$-dimensional space, with an effective dimension of $D$.
> We have provided the definitions in Appendix 6.6 and revised the first paragraph of Section 4.1 accordingly. The updated part has been highlighted for your reference.
>
> In addition to the subspace-based algorithms, we have included two recent advanced algorithm operating in the original space, , SAASBO and VanillaBO. We have reported their performance and provided comparisons with our approaches. We have also related them in Section 2 of Related Work. Thank you for your suggestions.

---

> > ### Comment · Reviewer_JcdP · 2025-03-18
> > **questions**
> >
> > Thank you for your response. Comparison with vanilla BO is informative. I have questions.
> >
> > - In the proof of (3), how (3) is derived from r^2(y_1,y_2) = (1 + O_p(d^-1/2)) r^2(x_1,x_2)?
> >
> > - The authors provided the error on kernel. Does that guarantee something about the error on 'f'?
> >
> > - In Fig.1, the definition of axis and color are ok. What is the 'surface' in the three plots? \cal{X}?

---

> > > ### Author Response · Authors · 2025-04-01
> > >
> > > Thank you so much for your additional comments, and apologies for the delayed response. I had assumed we couldn’t provide further replies.
> > >
> > > - many common kernels are smooth functions of $r^2(x,x')$. For example, the squared exponential kernel is$k(x,x')=\theta_0\exp\left(-r^2(x,x')/2\right)$. We will clarify this in the next revision.
> > >
> > > - In fact, the error does not directly lie in $f(x)$, but rather in its Gaussian process (GP) prior. However, the error in the kernel is valuable because Bayesian optimization utilizes the GP prior to update the GP fit. The function approximation is performed through GP fitting. Showing that the GP kernel can be well approximated by our CEP approach will demonstrate that the function approximation on f by the GP works effectively. We will clarify this in the next revision.
> > >
> > > - the function surface is represented by the color, with more yellow indicating larger values of f(x). But there might have been some confusion here. If so, please tell us know.

---

> > > > ### Comment · Reviewer_JcdP · 2025-04-01
> > > > **questions**
> > > >
> > > > Thank you for response but I still do not fully understand the answer.
> > > >
> > > > For the first question, I asked 'how (3) is derived from r^2(y_1,y_2) = (1 + O_p(d^-1/2)) r^2(x_1,x_2)?', but in my current understanding, the authors response is not rigid answer to this question.
> > > >
> > > > The last question is also not clear to me. In the left plot of figure 1, a 2dim nonlinear plane (which I called 'surface') is shown in three dimensional space. What is this? Note that I do not ask about the color.

---

> > > > > ### Author Response · Authors · 2025-04-02
> > > > >
> > > > > - on the equation (3)
> > > > >
> > > > > Let me explain it clearly here. In the revised version, we will add the details.
> > > > > Given the squared exponential kernel is $k(x,x')=\theta_0\exp\left(-r^2(x,x')/2\right)$,
> > > > > we have shown that $r^2(y,y')=(1+O_p(d^{-1/2}))r^2(x,x')/2$. Substituting this into the kernel follows
> > > > >  $$k(y,y')=\theta_0\exp\left(-r^2(y,y')/2\right)=\theta_0\exp\left(-(1+O_p(d^{-1/2}))r^2(x,x')/2\right).$$
> > > > > By applying a first-order Taylor expansion to $\exp\left(-(1+O_p(d^{-1/2}))r^2(x,x')/2\right)$ around $-r^2(x,x')/2$, it follows that (3) holds. This result can also be derived from the delta method.
> > > > >
> > > > > - on the plot
> > > > >
> > > > > Thank you for your patience regarding the plot. The confusion was due to my misunderstanding, but I now see the question clearly.
> > > > >
> > > > > The surface represents a slice within a cube. Since plotting all colors inside the cube would be too messy, we illustrate the concept by displaying only a slice and then showing the projections onto it. In notation, the slice is defined by an equation $g(x_1, x_2, x_3) = 0$, where the slice function serves an illustrative purpose.
> > > > >
> > > > > In the revised version, we will clarify this explanation. I sincerely appreciate your patience to point it out.

---

> > > > > > ### Comment · Reviewer_JcdP · 2025-04-02
> > > > > > **question**
> > > > > >
> > > > > > In my understanding, the Taylor expansion should be $k - O(d^{1/2}) r^2 k$, where $r = r(x,x')$ and $k = k(x,x')$. Why r^2 in the second term disappear in (3)?

---

> > > > > > > ### Author Response · Authors · 2025-04-02
> > > > > > >
> > > > > > > yes, you are right.  I just considered the case where $r^2$ is bounded, since $\exp(-r^2)\rightarrow 0$ as $r^2\rightarrow \infty$, making its contribution negligible in GP fit.
> > > > > > >
> > > > > > > Anyway, the equation (3) is not correct, so I truly appreciate your pointing that out.
> > > > > > >
> > > > > > > I will revise it accordingly.  When $r^2$ is bounded, (3) holds.  When $r^2=o(d^{1/2})$, $k(y,y')=(1+o(1))k(x,x')$. When $r^2/d^{1/2}$ is unbounded, $\exp(-r^2)=o(1)$ and $\exp(-r^2/d^{1/2})=o(1)$, following $k(x,x')=o(1)$ and $k(y,y')=k(x,x')+o(1)$. But when $r^2=O(d^{1/2})$, $k(y,y')=O(k(x,x'))$.  I will revise the kernel result and let it depend on $r^2/d^{1/2}$. I will make the corresponding revisions elsewhere. Thank you so much!

---

> > > > > > > > ### Comment · Reviewer_JcdP · 2025-04-02
> > > > > > > > **.**
> > > > > > > >
> > > > > > > > It seems somewhat odd to perform an asymptotic analysis with respect to $d$ while assuming that $r^2$ is bounded. Typically, we expect $D > d$, and $r^2$ appears to (linearly?) depend on $D$. I am also unsure whether it is appropriate to use a first-order Taylor expansion when $r^2$ becomes large. In the first place, a standard first-order Taylor expansion only provides an approximation (that is valid only for small $r^2$) and does not yield an exact equality. If higher-order terms are being absorbed into $O(d^{-1/2})$, more detailed explanations should have been provided, which may also be related to the bound or unbounded discussion.

---

> > > > > > > > > ### Author Response · Authors · 2025-04-03
> > > > > > > > >
> > > > > > > > > Thanks again! Your comments greatly improve this paper.
> > > > > > > > >
> > > > > > > > > To make things more concise, in the revised version, I will simply use the continuous mapping theorem.
> > > > > > > > >
> > > > > > > > > The revision is given as follows:
> > > > > > > > > From the continuous mappling theorem, we have $k(y,y')\rightarrow_p k(x,x')$ as $r^2(y,y')\rightarrow_p r^2(x,x')$.
> > > > > > > > >
> > > > > > > > > This version does not  explicitly quantify the approximation error, but it remains concise and serves our analytical goal: providing a good approximation.

---

### Review · Reviewer_Hm6j · 2025-02-23

**Summary Of Contributions:**

The authors claim 2 main contributions (1) use of different A_t in each iteration. (2) effective subspace assumption no longer needed. The second contribution is not clear to me which is why i have asked for clarity.

**Audience:**

Yes

**Claims And Evidence:**

Yes

**Requested Changes:**

More clarity and i have asked a couple of questions, please see the strengths and weaknesses section

**Strengths And Weaknesses:**

Review

The paper has some parts which are well-written, but misses out on explaining several important aspects that could have been much better explained for a reader not familiar with this subfield. The overall empirical results are promising, and I appreciate the authors sharing the code and important short comings of the proposed method in the conclusion section.


The main difference seems to be that the proposed algorithm samples a new matrix A_t in each iteration as opposed to the previous approaches. The authors have mentioned another improvement over previous works in the related works section -- that the acquisition function is evaluated to select the point in the embedded space by previous works while this paper does that in the original space. This second improvement is not clear at all. Algorithm 1, for example, calculates the acquisition criterion in the embedded space which is just projected back to the original space. This can be done for all the previous approaches too, so where exactly is the new contribution and how does this not assume the existence of an effective subspace?

The authors could expand the discussion on acquisition function, taking a couple of examples and its importance. For the numerical results using various benchmark functions (Holder, Griewank etc) also please provide more clarity on the functional forms of these functions and their importance.

What is your intuition about why the method performs well for noisy data?

Why are the results for many real world datasets for small d only ?

Minor:

Typo pg 4 "difference between x^\tilde{x}... should be \tilde{x}^\tilde{x}

---

> ### Author Response · Authors · 2025-03-15
>
> Thank you for numerous constructive comments to enhance our manuscript. We have carefully addressed each comment. Below we describe the changes among them.
>
> - On the analysis
>
> In the previous version, we only provided the concentration of the transformed $\tilde{x}$ to the original $x$, which was indeed insufficient. In this version, we have presented an analysis of the approximate error of the GP fit to show that the CEP approach preserves the variance function of a Gaussian process. The new analysis also explains why CEP approach does not rely on the effective subspace assumption.
>
> - On the ``second improvement" over previous ones
>
> We do calculate the acquisition criterion in the embedded space, which is then projected back to the original space.
> However, the selected point lies in the embedded space $\mathcal{Y}$, while the function $f()$ is evaluated in the original space $\mathcal{X}$. The expansion projection ensures that the fit based on transformed $\tilde{x}$ closely approximates that on the original $x$.
>
> - On the intuition about why the method performs well for noisy data
>
> From the analysis provided in this revision, the CEP approach preserves the variance function of a Gaussian process, which ensures robustness to data noise, as it does not explicitly depend on the noise.
>
> - On  the small $d$ only for many real world datasets
>
> In the four real-world datasets with $D = 12, 14, 36, 60$, we evaluate the performance for $d = 2$ and $d = 5$ as illustrative examples, consistent with the simulation results showing that these values work well. In the conclusion, we have included a discussion on the practical selection of $d$.

---

### Comment · Action_Editor_H2Rk · 2025-02-24
**3 Reviews are in now**

Dear authors,

3 reviews are in now.
Specific questions and change requests were provided.
Please carefully read them and send us your rebuttal.

--AE

---

### Author Response · Authors · 2025-03-15

Thank you for the reviewers' constructive comments, which have helped improve our manuscript much.
We also appreciate the AE for granting an additional week for this revision.  We have carefully addressed each comment.
The main changes are as follows:

- On the analysis of the CEP approach

In the previous version, we only provided the concentration of the transformed $\tilde{x}$ to the original $x$, which was indeed insufficient. In this version, we have presented an analysis of the approximate error of the GP fit to show that the CEP approach preserves the variance function of a Gaussian process.
The new analysis also explains why CEP approach does not rely on the effective subspace assumption.

- On Figure 1 of the illustration of CEP

In these plots, the colors represents functions values. We have re-plotted this figure to improve clarity.

- On the experiments

In addition to the subspace-based algorithms, we have included two recent advanced algorithms operating in the original
space, SAASBO and VanillaBO.

---

### Decision · Action_Editor_H2Rk · 2025-04-09

**Recommendation:** Reject

**Comment:**

The primary innovation of this work is the elimination of the effective dimension assumption in Bayesian optimization. However, this assertion has not been fully supported in the current manuscript, particularly due to the limitations of the probabilistic asymptotic analysis highlighted by Reviewer JcdP. Furthermore, Reviewer mE4T has noted that the experimental evaluation provided is inadequate to validate the utility of the proposed algorithm. As a result, the current submission cannot be accepted for publication at this time.

Should the authors undertake a major revision that effectively addresses both the theoretical issues identified in the analysis and those arising from the empirical evaluation, they may wish to resubmit a fully revised manuscript at a later date.

**Audience:**

High-dimensional Bayesian optimization is a topic of broad interest among researchers in machine learning methodology and AI for Science. Consequently, the potential audience for this field is vast.

**Claims And Evidence:**

Thank you for your rebuttal and revised manuscript.

While some of the reviewers' concerns have been adequately addressed and certain unclear points have been clarified, they still have reservations about the current manuscript. Specifically, the manuscript has been criticized for its probabilistic asymptotic analysis, imprecise technical descriptions, and a deficiency in empirical evidence. These are considered critical issues that need to be addressed to substantiate the validity of the work.

**Resubmission Of Major Revision:**

The authors may consider submitting a major revision at a later time.